# MICU1 controls spatial membrane potential gradients and guides Ca$^{2+}$ fluxes within mitochondrial substructures

Benjamin Gottschalk [1], Zhanat Koshenov [1], Markus Waldeck-Weiermair[1], Snježana Radulović[2], Furkan E. Oflaz [1], Martin Hirtl [1], Olaf A. Bachkoenig[1], Gerd Leitinger [2], Roland Malli [1,3] & Wolfgang F. Graier [1,3✉]

Mitochondrial ultrastructure represents a pinnacle of form and function, with the inner mitochondrial membrane (IMM) forming isolated pockets of cristae membrane (CM), separated from the inner-boundary membrane (IBM) by cristae junctions (CJ). Applying structured illumination and electron microscopy, a novel and fundamental function of MICU1 in mediating Ca$^{2+}$ control over spatial membrane potential gradients (SMPGs) between CM and IMS was identified. We unveiled alterations of SMPGs by transient CJ openings when Ca$^{2+}$ binds to MICU1 resulting in spatial cristae depolarization. This Ca$^{2+}$/MICU1-mediated plasticity of the CJ further provides the mechanistic bedrock of the biphasic mitochondrial Ca$^{2+}$ uptake kinetics via the mitochondrial Ca$^{2+}$ uniporter (MCU) during intracellular Ca$^{2+}$ release: *Initially*, high Ca$^{2+}$ opens CJ via Ca$^{2+}$/MICU1 and allows instant Ca$^{2+}$ uptake across the CM through constantly active MCU. *Second*, MCU disseminates into the IBM, thus establishing Ca$^{2+}$ uptake across the IBM that circumvents the CM. Under the condition of MICU1 methylation by PRMT1 in aging or cancer, UCP2 that binds to methylated MICU1 destabilizes CJ, disrupts SMPGs, and facilitates fast Ca$^{2+}$ uptake via the CM.

[1] Gottfried Schatz Research Center: Molecular Biology and Biochemistry, Medical University of Graz, Graz, Austria. [2] Gottfried Schatz Research Center: Cell Biology, Histology and Embryology, Medical University of Graz, Graz, Austria. [3] BioTechMed Graz, Graz, Austria. ✉email: wolfgang.graier@medunigraz.at

The ultrastructure of the inner mitochondrial membrane (IMM) is separated into two parts, the inner boundary membrane (IBM) representing the interface of the IMM with the outer mitochondrial membrane (OMM), and the cristae membrane (CM), forming the membrane invaginations into the mitochondrial matrix[1]. Morphologically, the cristae junctions (CJ), bottleneck membrane structures with a diameter of approximately 17 nm[2,3], separate IBM and CM from each other. The mitochondrial contact site and cristae organizing system (MICOS) proteins and the scaffold protein optic atrophy 1 (OPA1) stabilize the CJ[4].

All three IMM compartments CJ, IBM, and CM differ in morphology and composition of characteristic proteins according to their spatio-specific functions. In particular, the IBM hosts the mitochondrial protein uptake complex TIM (translocase of the inner membrane) that interacts with the TOM (translocase of the outer membrane) complex in the OMM to translocate nuclear-encoded mitochondrial proteins into the mitochondrial matrix or mitochondrial membranes[5]. The CM contains the four complexes of the respiratory chain[6]. Of those, complex I, III, and IV exhibit proton pump activity and, thus, contribute to the mitochondrial membrane potential ($\Delta\Psi_m$) across the IMM. The $\Delta\Psi_m$ represents the energetic force for the production of ATP by the $F_0F_1$ ATPase that is also localized to the CM[7]. By knowing the spatial restriction of complexes I, III and IV to the CM, experimentally determined differences in pH of CM (pH 7.0)[8] and mathematically estimated pH values at the IBM (pH 7.4)[9] become comprehensible. The CJ, representing a physical diffusion barrier, constrains the diffusion of the protons to the cristae lumen. Rieger et al. showed that a pH gradient is even established between the Complex IV and the $F_0F_1$ ATPase in the isolated CM compartment[8]. The proton motive force towards the matrix is the main component contributing to establish the $\Delta\Psi_m$[10]. Wolf et al. showed recently that individual cristae are isolated insulators restricting the membrane potential generated at the CM by the CJ. This study demonstrated two distinct membrane potentials at the CM ($\Delta\Psi_{CM}$) and the IBM ($\Delta\Psi_{IBM}$)[11].

Mitochondrial $Ca^{2+}$ uptake, which highly depends on $\Delta\Psi$, is essential to activate $Ca^{2+}$-dependent dehydrogenases of the TCA cycle and increase oxidative phosphorylation[12]. Mitochondrial $Ca^{2+}$ Uniporter (MCU) complex-mediated $Ca^{2+}$ uptake is driven by the $\Delta\Psi_m$ across the IMM, yet it was never elaborated if the different CM ($\Delta\Psi_{CM}$) and the IBM ($\Delta\Psi_{IBM}$) potentials influence the $Ca^{2+}$ uptake on the level of mitochondrial ultrastructure.

The main regulator of MCU, mitochondrial $Ca^{2+}$ uptake 1 (MICU1), is localized at the IBM, stabilizes the CJ, and interacts with the MICOS complex[2,13]. Under basal conditions, MICU1 exists as a hexamer or oligomer, which in case of $Ca^{2+}$ binding via EF-hands of MICU1 disassembles into dimers with an $EC_{50}$ of 3.8 µM $Ca^{2+}$[14,15]. Methylation of MICU1 by protein arginine methyltransferase 1 (PRMT1) was shown to reduce the $Ca^{2+}$-binding affinity of MICU1 to approximately 18.5 µM and inhibit thereby the $Ca^{2+}$ induced dissociation into dimers. Contrary, the specific binding of uncoupling protein 2 (UCP2) to methylated MICU1 counteracts the methylation effect and resensitizes MICU1 towards $Ca^{2+}$ ($EC_{50} = 4.0$)[15]. Intracellular ER $Ca^{2+}$ release from the endoplasmic reticulum (ER) leads to OMM located $Ca^{2+}$ hotspots (up to 16.42 µM) in close proximity to mitochondrial-associated ER membranes (MAM), while mitochondrial regions without direct contact with the ER are facing much lower $Ca^{2+}$ concentration of approximately 2.94 µM[16]. The precise regulation of MICU1 by PRMT1 and UCP2 is important for cell metabolism and viability[17]. Especially cancer cells show a clear increase in UCP2 but also PRMT1 expression and both proteins when highly expressed are prognostic markers for lung carcinoma patients[18].

The ability of MICU1 to respond to $Ca^{2+}$ hotspots with changes in its quaternary structure paired with the observation that MICU1 stabilizes CJ led us to hypothesize that the opening and stability of the CJ are directly regulated by high $Ca^{2+}$[19]. Strong opening of the CJ is associated with a loss of membrane potential[2,11], impaired oxidative phosphorylation and ATP production[20,21], and induction of apoptosis[22]. Accordingly, we used structured illumination microscopy, differential membrane potential measurements of the CM and IBM, electron microscopy, and mitochondria-targeted genetically encoded biosensors for $Ca^{2+}$ to elaborate the involvement of MICU1 in CJ dynamics, effects on mitochondrial $Ca^{2+}$ uptake, and $\Delta\Psi_m$ homeostasis between IBM and CM.

## Results

**Spatial membrane potential distributions exist between CM and IBM**. To detect spatial membrane potential gradients (SMPGs) between the CM ($\Delta\Psi_{CM}$) and the IBM ($\Delta\Psi_{IBM}$), we established a reliable and robust method using super-resolution microscopy. This approach builds on the assumption that the fluorescent potentiometric dye tetramethylrhodamine methyl ester (TMRM) is incorporated into the IMM[23–25] according to local $\Delta\Psi$, thus, the individual fluorescent intensities reflect distinct $\Delta\Psi_{IBM}$ and $\Delta\Psi_{CM}$. While at low and moderate concentrations TMRM binds to the IMM in a linear correlation with the $\Delta\Psi_m$, saturation effects have been reported at high TMRM concentrations[21]. Further, we could show that TMRM concentrations in the range of 1.35–81 nM show a slight saturation effect (Supplementary Fig. 1). Therefore, we assumed that at a given high concentration TMRM gets saturated in the CM while the TMRM accumulation maintains in a linear correlation in the IBM with its lower $\Delta\Psi$. Thus, we hypothesized that both sub-mitochondrial compartments vary in their labeling properties and the shifted TMRM affinity can be used to distinguish the different membrane potentials of the two sub-compartments of the IMM.

At very high concentrations of TMRM, self-quenching effects are known[25]. Therefore, we measured concentrations of 1.35, 13.5, 81, 200, 500, and 1000 nM TMRM for self-quenching effects. For 1.35–81 nM no self-quenching effects were observed and only at 500–1000 nM TMRM we measured considerable self-quenching (Supplementary Fig. 2), as reported elsewhere[26,27]. Accordingly, the used range of 1.35–81 nM TMRM used to detect mitochondrial membrane potential gradients was not affected by TMRM self-quenching effects.

To ensure that TMRM is specifically labeling and accumulating in the IMM, HeLa cells expressing mitochondrial matrix targeted superfolder green fluorescent protein (mt-sfGFP) and stained with 81 nM TMRM were treated with 100 µM ATP to induce mitochondrial swelling. A clear separation of TMRM cristae labeling and matrix mt-sfGFP fluorescence became obvious, showing specific labeling of the IMM by TMRM (Supplementary Fig. 3), as shown by others[23,24].

In a first attempt, we investigated the sub-mitochondrial localization of TMRM and mt-sfGFP using super-resolution dual-color structured illumination microscopy (SIM). At low TMRM concentrations (≤5.4 nM) the overlay of TMRM with mt-sfGFP shows an exclusive TMRM staining of the cristae membrane (Fig. 1a). At TMRM concentrations ≥13.5 nM an additional halo of TMRM fluorescence surrounding the matrix develops (Fig. 1a), pointing to stronger labeling of the inner boundary membrane (IBM).

Two different methods were used to quantify the spatial distribution of TMRM in relation to the reference channel mt-sfGFP: (1) The full width of half maxima (FWHM) of mitochondria for the TMRM and mt-sfGFP (i.e., matrix) channels were measured

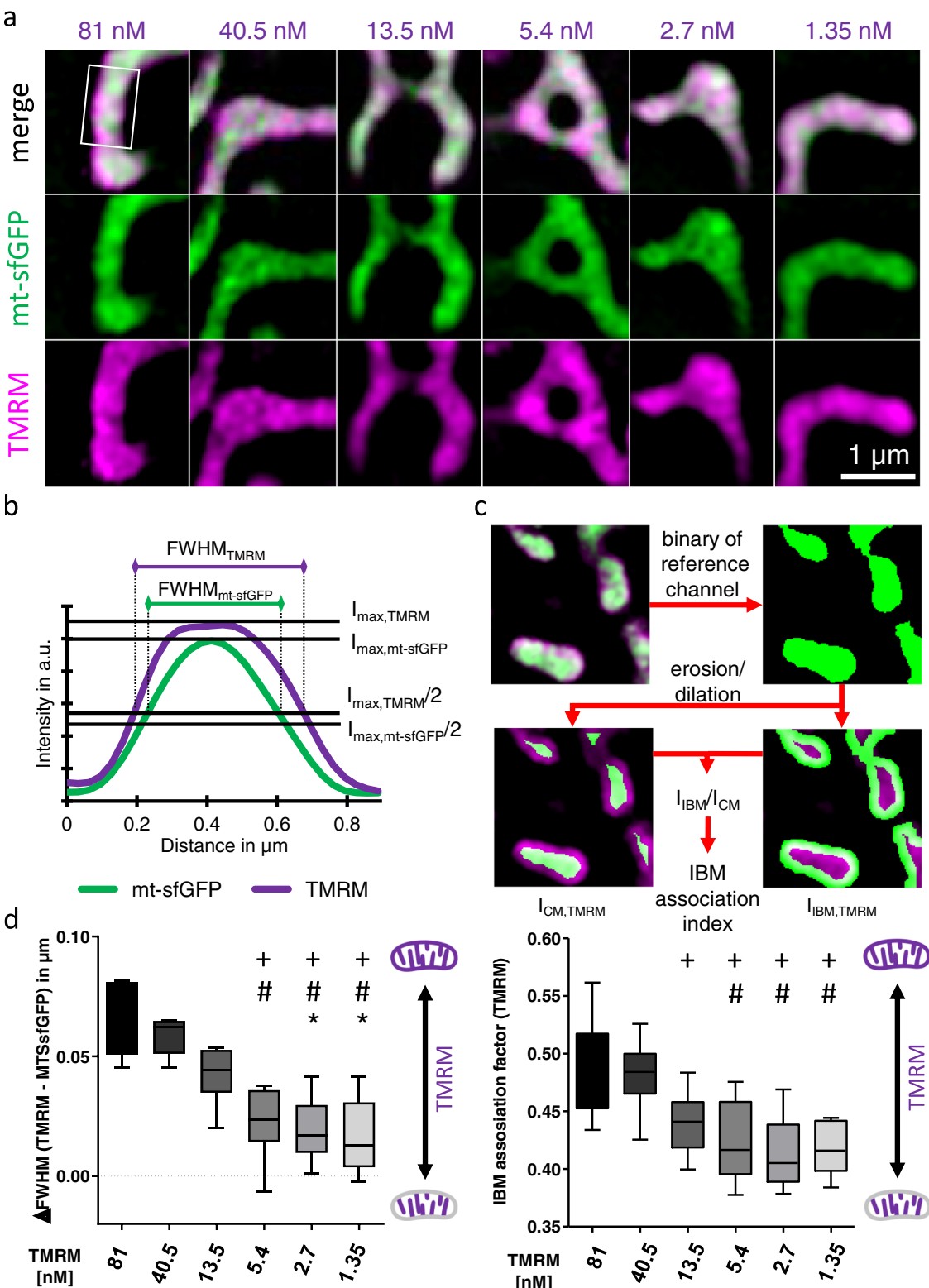

and their differences ($\Delta FWHM_{(TMRM - mt\text{-}sfGFP)}$) illustrate the concentration-dependent TMRM expansion (Fig. 1b). (2) The IBM association index ($AI_{IBM}$)[2], which correlates the TMRM fluorescence intensities of the central cristae membrane (CM) with the IBM (Fig. 1c). For both parameters, $\Delta FWHM_{(TMRM - mt\text{-}sfGFP)}$ and $AI_{IBM}$, a sigmoidal correlation between a TMRM accumulation in the mitochondrial periphery (halo) and increasing TMRM concentration was found, indicating different membrane potentials at the CM and IBM (Fig. 1d).

**MICU1 and OPA1 contribute to the isolation of CM and IBM potentials.** The CJ represents the bottleneck structure separating CM and IBM, thereby enabling high proton concentration gradients across the CM[11]. MICU1[2] and OPA1[4,28] are known to contribute directly to the stability of the CJ. Accordingly, the contribution of these proteins to the isolation of the different mitochondrial membrane potentials ($\Delta\Psi_m$) in CM ($\Delta\Psi_{CM}$) and IBM ($\Delta\Psi_{IBM}$) was investigated. Knockdown of either MICU1[29] (Supplementary Fig. 4) or OPA1[30] leads to an increase in IBM vs.

**Fig. 1 Measuring cristae and inner boundary membrane potential gradient. a** Representative images of HeLa cells expressing mt-sfGFP (green), stained with 81, 40.5, 13.5, 5.4, 2.7, or 1.35 nM tetramethylrhodamine methyl ester (TMRM) (magenta) and examined using simultaneous dual-color 3D-SIM. **b** Intensity line plots of the mitochondrion in the white box in **a** showing the mt-sfGFP and TMRM intensity distribution. Full width at half maximum (FWHM) of mt-sfGFP and TMRM distributions are indicated with the corresponding maximal ($I_{max}$) and half-maximal ($I_{max}/2$) intensities. **c** Schematic illustration of the calculation of the IBM association index on the example of HeLa cells expressing mt-sfGFP (green) and stained with 81 nM TMRM (magenta). The reference channel (mt-sfGFP) is thresholded and split by erosion and dilation into inner boundary membrane (IBM) and cristae membrane (CM) related segments, which are used as masks to measure TMRM mean intensities. The ratio of IBM intensity ($I_{IBM,TMRM}$) and CM intensity ($I_{CM,TMRM}$) results in the IBM association index. **d** Quantitative analysis of **a** using the ΔFWHM of TMRM and mt-sfGFP (described in **b**) or the IBM association index (described in **c**) to determine TMRM distribution to the IBM. The higher ΔFWHM or IBM association index, the broader the TMRM distribution indicating a stronger TMRM staining in the IBM. Data information: Horizontal lines in **d** represent the median, the lower and upper hinge show, respectively, first quartile and third quartile, and lower and upper whiskers encompass minimal and maximal values. Images and analyses were obtained from each 9–10 cells in 6 independent experimental days ($n = 6$). *$P < 0.05$ vs. 13.5 nM, #$P < 0.05$ vs. 40.5 nM and +$P < 0.05$ vs. 81 nM TMRM conditions evaluated using one-way analysis of variance (ANOVA) with Bonferroni post hoc test.

CM staining at low TMRM concentrations (i.e. 13.5, 5.4 nM) (Fig. 2a–c and Supplementary Fig. 5), thus, indicating that an intact CJ essentially needs either MICU1 or OPA1 and is fundamental to establish a high $ΔΨ_m$ restricted to the cristae. Investigations of the mitochondrial morphology revealed smaller and rounded mitochondria with lower interconnectivity in MICU1 and OPA1 depleted cells (Supplementary Fig. 6). No mitochondrial swelling was observed upon depletion of either MICU1 or OPA1 and TMRM did not affect the mitochondrial morphology (Supplementary Fig. 7).

Because UCP2 was recently shown to interact with methylated MICU1 and, thereby, pseudo-normalizing the $Ca^{2+}$ sensitivity and $Ca^{2+}$ dependent quaternary structure of methylated MICU1[15], we next tested the roles of UCP2 and PRMT1 in CJ stability. As indicated by the low $ΔFWHM_{(TMRM - mt-sfGFP)}$ and $AI_{IBM}$, knockdown of UCP2[18,31] results in the accumulation of TMRM in the CM for all TMRM concentrations applied (Fig. 2a–c and Supplementary Figs. 5 and 6). These data indicate that the lack of UCP2 yields a hyperpolarization of the CM vs. IBM. Based on our previous observations[2,15] it is tempting to speculate that UCP2 knockdown strengthens CM isolation from the IBM, since UCP2 does not interfere with methylated MICU1. This assumption is further supported by our findings that double knockdown of UCP2 and MICU1 prevented CM hyperpolarization upon UCP2 depletion (Fig. 2a–c and Supplementary Figs. 5 and 6), thus, pointing to a MICU1-dependent role of UCP2.

Next, we tested the influence of PRMT1 silencing[18] on the $ΔΨ_m$ distribution. PRMT1 knockdown lead to a clear clustering of ΔΨ at the CM (Fig. 2a–c and Supplementary Figs. 5 and 6), confirming the exclusive function of UCP2 on methylated MICU1. Accordingly, experimental results from cells that were depleted from UCP2 and PRMT1 were not different from a single knockdown experiment (Fig. 2a–c and Supplementary Figs. 5 and 6). Structurally, knockdown of UCP2 and PRMT1 lead to higher mitochondrial interconnectivity and elongated shape while mitochondrial thickness did not substantially change (Supplementary Fig. 7).

**MICU1, UCP2, and OPA1 knockdown influences spatial CM distribution.** Recently, we reported that a knockdown of either MICU1 or OPA1 enlarge CJ width and affect cristae morphology[2]. To verify whether changes in IBM/CM TMRM ratio manifest due to the morphological changes in the IMM structure, the distribution of CM across the entire mitochondrial diameter was evaluated in MICU1, OPA1 or UCP2 depleted cells with transmission electron microscopy (Fig. 3a). MICU1 or OPA1 knockdown did not affect the ratio of the OMM to CM length (Fig. 3b). However, normalizing the cristae perimeter to the mitochondrial area, unveiled increased cristae density in OPA1 knockdown cells (Fig. 3c). OPA1 and MICU1 knockdown

lead to mitochondrial fragmentation (Supplementary Fig. 7) while the relative amount of CM in the mitochondria remains unchanged (Fig. 3b). The increased cristae density in OPA1 and MICU1 knockdown cells most likely relates to the decreased mitochondrial area/perimeter ratio. Silencing of UCP2 did not change cristae density.

However, the IBM/CM TMRM intensity ratios may also be influenced by the spatial distribution of CM within mitochondria. To evaluate this possibility, consecutive circular segments of mitochondria (shown in Fig. 3d) were analyzed in regard to the cristae density ($ρ_{CM}$). Both, MICU1 and OPA1 depletion result in a focused elevation of $ρ_{CM}$ in the mitochondrial center (Fig. 3e, f). Despite central clustering of IMM upon depletion of OPA1 or MICU1, both knockdowns lead to higher $ΔFWHM_{(TMRM - mt-sfGFP)}$ and IBM association index (Fig. 2b, c). Because direct adaption of spatial membrane quantity would have led to increased CM fluorescence intensity under MICU1 or OPA1 knockdown, these results indicate that the spatial $ρ_{CM}$ distribution does not disturb the super-resolution measurements of $ΔΨ_{IBM}/ΔΨ_{CM}$. The knockdown of UCP2 causes a strong increase of $ρ_{CM}$ in the mitochondrial periphery (Fig. 3e, f), most likely derived from an increased number of CJ (Supplementary Fig. 8).

Additionally, we evaluated whether or not the addition of TMRM in the various concentrations influences the cristae morphology or density. Neither the amount of CM nor $ρ_{CM}$ was changed while the spatial distribution of $ρ_{CM}$ was slightly elevated in the mitochondrial center with increased TMRM concentrations (Supplementary Fig. 9).

**$Ca^{2+}$ activation of MICU1 triggers a transient CJ opening and spatial cristae depolarization.** MICU1 contains two EF-hands and undergoes structural rearrangements upon $Ca^{2+}$ binding[14]. Accordingly, we investigated the role of MICU1 in the opening of CJ upon $Ca^{2+}$ mobilization. Therefore, HeLa cells expressing an ER marker were stained with TMRM (5.4 nM), and the respective fluorescence intensities and distributions were recorded upon stimulation with histamine (Fig. 4a), a potent IP$_3$-generating agonist. In this experimental setting, we were able to measure local variations of $ΔΨ_m$ depending on the proximity to the ER.

These experiments revealed that upon histamine stimulation HeLa cells react with local restricted losses of $ΔΨ_m$ in IMM regions with close proximity to MAMs and punctually along the entire IMM (Fig. 4b). The local TMRM spikes with a duration of several seconds indicate restricted fluctuations of the membrane potential of single cristae in close temporal correlation to cytosolic $Ca^{2+}$ elevations that trigger structural rearrangement of MICU1, destabilize the CJ and lead to instant local equilibration of $ΔΨ_{CM}$ and $ΔΨ_{IBM}$.

We have recently shown that silencing of MICU1 destabilizes the CJ[2] and found herein that MICU1 knockdown leads to a

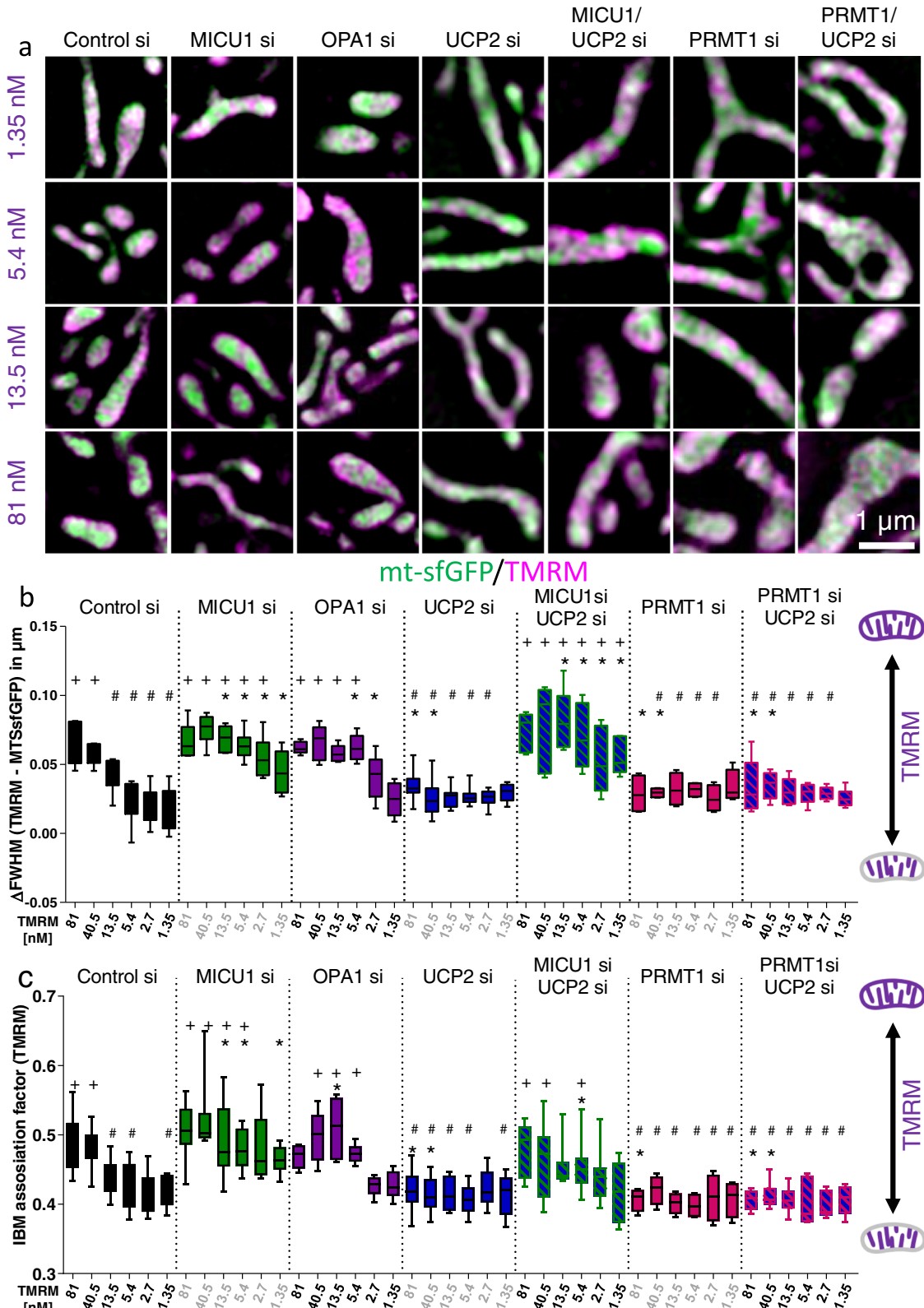

homogenization of $\Delta\Psi_{CM}$ and $\Delta\Psi_{IBM}$ (Fig. 2a–c). As expected, siRNA-mediated knockdown of MICU1, and by this means absence of SMPGs, resulted in a loss of histamine-induced spatial $-\Delta/+\Delta$TMRM peaks (Fig. 4c).

**The role of UCP2 and Ca$^{2+}$ hotspots on CJ integrity/stability.** We previously reported that UCP2 normalizes the reduced Ca$^{2+}$

sensitivity of PRMT1 methylated MICU1 and, thus, decreases its ability for Ca$^{2+}$-induced fragmentation[14,15]. Knockdown of UCP2 in HeLa cells with high PRMT1 activity decreased the overall $-\Delta/+\Delta$TMRM response to histamine but maintained the amounts of spatial TMRM drops near MAMs (Fig. 4d). The reduced overall $-\Delta/+\Delta$TMRM ratio implies that the absence of UCP2 leads to a desensitization of MICU1 to Ca$^{2+}$.

**Fig. 2 Cristae and inner boundary membrane potential gradient are controlled by MICU1, UCP2, and OPA1. a** Representative images of HeLa cells expressing mt-sfGFP (green) and single or double transfected with Control, MICU1, OPA1, UCP2, or PRMT1 siRNA, stained with 81, 13.5, 5.4, or 1.35 nM tetramethylrhodamine methyl ester (TMRM) (magenta) and examined using simultaneous dual-color 3D-SIM. **b** Quantitative analysis of TMRM concentrations presented in **a** plus 40.5 and 2.7 nM TMRM concentration using the ΔFWHM of TMRM and mt-sfGFP to determine TMRM association with the IBM. The higher ΔFWHM, the broader the TMRM distribution indicating a stronger TMRM staining in the IBM. **c** Quantitative analysis of the same TMRM concentrations used in **b** using the IBM association factor of TMRM to determine TMRM association with the IBM. The relation of the IBM association factor to the TMRM distribution is similar to that of ΔFWHM **b**. Data information: horizontal lines represent the median, the lower and upper hinge show respectively first quartile and third quartile, and lower and upper whiskers encompass minimal and maximal values. Images and analyses were obtained from each 9-10 cells in 5-10 independent experimental days ($n_{Control\ si} = 8$, $n_{MICU1\ si} = 7$, $n_{OPA1\ si} = 5$, $n_{UCP2\ si} = 10$, $n_{MICU1\ si\ /UCP2\ si} = 7$, $n_{PRMT1\ si} = 6$, $n_{PRMT1\ si\ /\ UCP2\ si} = 6$). *$P < 0.05$ vs. respective control si #$P < 0.05$ vs. respective MICU1 si and +$P < 0.05$ vs. respective UCP2 si conditions carried out with two-way analysis of variance (ANOVA) and Bonferroni post hoc test.

Consequently, we conclude that in UCP2-depleted HeLa cells the CJ does not undergo conformational changes upon histamine stimulation, while the $Ca^{2+}$ hotspots in the MAM region are high enough to force MICU1 deoligomerization leading to an opening of the CJ and thereby homogenizes IBM and CM potential.

Control experiments adding $Ca^{2+}$ buffer (CaB) instead of histamine showed no response in $-\Delta/+\Delta$TMRM and basal $-\Delta/+\Delta$TMRM ratio did not change by silencing either MICU1 or UCP2 compared to control cells (Fig. 4c, d).

Store-operated $Ca^{2+}$ entry (SOCE) after ER $Ca^{2+}$ depletion, which leads to a homogeneous cytosolic $Ca^{2+}$ elevation, was used to analyze $-\Delta/+\Delta$TMRM alterations (Supplementary Fig. 10). After $Ca^{2+}$ re-addition no noticeable signals in $-\Delta/+\Delta$TMRM were detected for Control, MICU1, or UCP2 siRNA treated cells, supporting our assumption that only high local $Ca^{2+}$ concentration can lead to the opening of the CJ and spatial depolarization of the CM (Fig. 4d).

**Spatial intensity of cytosolic $Ca^{2+}$ determines subcompartmental $Ca^{2+}$ propagation in mitochondria.** The local depolarization of CM at MAMs upon stimulation with an IP₃-generating agonist points to a CJ opening and permeability for ions into the cristae lumen (CL). Therefore, we tested the subcompartmental $Ca^{2+}$ kinetics in mitochondria upon IP₃-induced ER $Ca^{2+}$ release and SOCE in MICU1, and UCP2 siRNA treated cells using red and green fluorescent genetically encoded $Ca^{2+}$ biosensors targeted either to the IMS and mitochondrial matrix, CL and mitochondrial matrix, or IMS and CL, respectively. The temporal delay of $Ca^{2+}$ propagation within the respective subcompartment of mitochondria was quantified as the delta of areas under the curve (ΔAUC) for normalized IP₃- and SOCE-induced $Ca^{2+}$ signals in the respective sub-compartment (Fig. 5a–c).

By silencing MICU1, the delay between the $Ca^{2+}$ signal in the IMS and those in the CL and the matrix were reduced in SOCE-but not IP₃-induced signals (Fig. 5d, e). The $Ca^{2+}$ propagation from the CL into the matrix was not affected during SOCE (Fig. 5f). Thus, the $Ca^{2+}$ propagation from the IMS to the cristae is under the control of MICU1 and the rate-limiting phase of the $Ca^{2+}$ propagation during SOCE. Nevertheless, the increased CL and decreased matrix $Ca^{2+}$ concentrations under SOCE in MICU1 knockdown cells indicate that MICU1 not only guards the CJ but also activates MCU-mediated $Ca^{2+}$ influx into the matrix (Supplementary Fig. 11). Moreover, while the IP₃-induced IMS to CL $Ca^{2+}$-kinetics remained unaffected, the IMS to matrix and CL to matrix $Ca^{2+}$ kinetics were clearly accelerated by MICU1 silencing (Fig. 5d–f and Supplementary Fig. 12).

Silencing of UCP2 leads to a delayed propagation of $Ca^{2+}$ signals from the IMS to the CL and matrix after IP₃-induced ER $Ca^{2+}$ release. In contrast, the $Ca^{2+}$ transition from the CL to the matrix was not affected (Fig. 5g–i and Supplementary Fig. 12). Furthermore, CL and matrix $Ca^{2+}$ delta intensities were reduced under the knockdown of UCP2 (Supplementary Fig. 11),

indicating a blockage of the CJ by UCP2 knockdown. For SOCE-induced mitochondrial $Ca^{2+}$ uptake, UCP2 silencing did not affect the $Ca^{2+}$ propagation between the various mitochondrial sub-compartments (Fig. 5g–i and Supplementary Fig. 12).

Taken together, these data indicate a biphasic $Ca^{2+}$ uptake mechanism that is dependent on the origin and spatial intensity of the cytosolic $Ca^{2+}$ signal and the stability of the CJ. In the first phase, IP₃-induced ER $Ca^{2+}$ release leads to cytosolic $Ca^{2+}$ hotspot in the MAMs[16]. Subsequently, MICU1 deoligomerizes upon $Ca^{2+}$ binding at these hotspots leading to spatial CL depolarization due to a $Ca^{2+}$ induced opening of CJs (Fig. 4b) and consequently to the flooding of the CL with $Ca^{2+}$, which, in turn, is instantly taken up into the matrix by constitutively active MCU within the CM. As $Ca^{2+}$ hotspots[16] and CJ openings (Fig. 4b) are short-lived, the second phase is dominated by the slower MICU1 mediated MCU shuttling into the IBM[2] achieving the second phase of $Ca^{2+}$ uptake into the mitochondrial matrix excluding the CL from the $Ca^{2+}$ uptake process (Fig. 5j). Notably, Cytosolic $Ca^{2+}$ signals originating from SOCE do not open the CJ due to a lack of $Ca^{2+}$ hotspots. Thus, the propagation of SOCE signals into the mitochondrial matrix relays only on the MCU-shuttling due to partially deoligomerized MICU1 similar to the second phase in IP₃-induced ER-$Ca^{2+}$ release (Fig. 5k).

## Discussion

In this study, we combined dual-color SIM, TEM, and sophisticated live-cell imaging with mitochondrial sub-compartmental targeted genetically encoded biosensors to investigate the fundamental function of MICU1, OPA1, and UCP2 in the dynamic regulation of spatial membrane potential gradients (SMPGs) between IBM and CM. Moreover, the individual role of the two distinct IMM compartments (i.e. IBM and CM) in the orchestration of mitochondrial $Ca^{2+}$ uptake has been elucidated. The CM and the IBM are isolated by the CJ that consists of the "mitochondrial contact site and cristae organizing system" (MICOS)[28,32] the optic atrophy 1 (OPA1)[28,32] and MICU1[2,13].

Confirming recent results from ref. [11], we herein demonstrate that the membrane potential of the IMM ($\Delta\Psi_{IMM}$) is separated by the CJ into two distinct potentials, that of the IBM ($\Delta\Psi_{IBM}$) and the CM ($\Delta\Psi_{CM}$). Notably, our data show that silencing of MICU1 leads to a strong shift of $\Delta\Psi_{IBM}/\Delta\Psi_{CM}$ towards the IBM. This disruption of SMPGs points to a fundamental role of MICU1 in CJ integrity, thus, reemphasizing our previous results showing a CJ opening and clear reduction of the overall $\Delta\Psi_{IMM}$ for MICU1 knockdown conditions[2]. Moreover, our new data point to an alternative model of mitochondrial uncoupling that builds on the erosion of SMPGs by means of CJ-controlled $\Delta\Psi_{CM}$. Opening of the CJ and the dissemination of the protons from the CL into the IMS yields depolarization of $\Delta\Psi_{CM}$ that mimics a mitochondrial uncoupling across the IMM.

We found in our recent study that the CJ widened through silencing of OPA1 or MICU1[2]. We assume that a destabilization

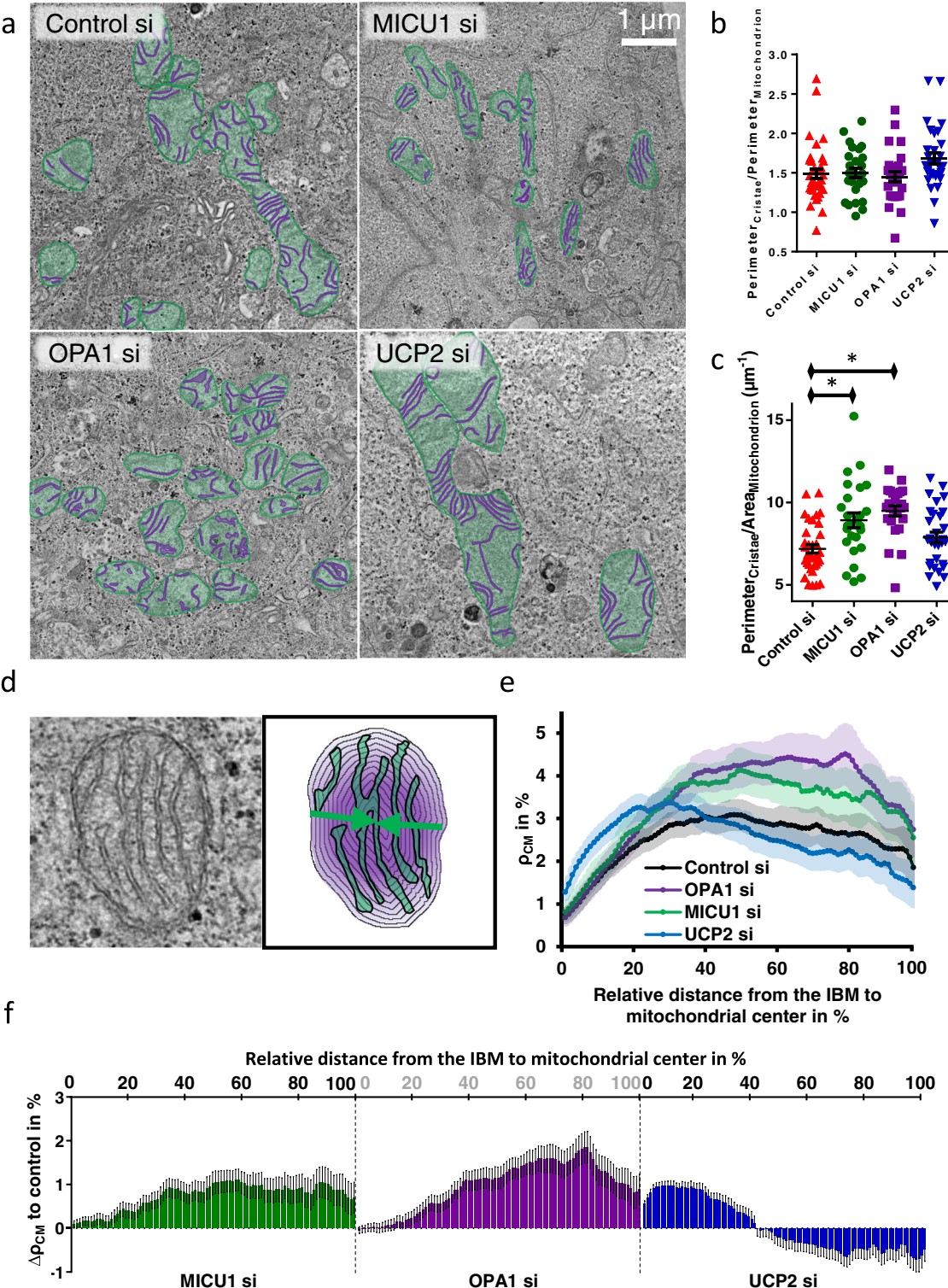

of the CJ leads to IMM constrictions which were shown to precede mitochondrial fission in ER close regions[19,33]. Our findings in this study suggest that mitochondrial fragmentation observed in MICU1 and OPA1 silenced cells is caused by destabilized CJ in ER close regions leading to IMM constriction and mitochondrial fission. Contrary reduction of UCP2 leads to a more stable CJ configuration[19] and thus to elongated mitochondria.

Because silencing of MICU1 destabilizes CJ under basal conditions and yields disruption of SMPGs, we speculate that even under resting condition, basal ER-originated $Ca^{2+}$ cycling within the MAMs achieves rare activation of MICU1 and, thus, increase in $H^+$-permeability of the CJ. Besides the process of $Ca^{2+}$ cycling, CJ $H^+$-permeability is further influenced by the methylation status of MICU1 by PRMT1. Methylation of MICU1 lowers the $Ca^{2+}$ binding affinity of MICU1 ($EC_{50}$ for $Ca^{2+}$ increases from 3.8 to 18.5 μM)[16] resulting in an enhanced stability/impermeability of the CJ. However, UCP2 binds exclusively to methylated MICU1, restores $Ca^{2+}$ sensitivity of methylated MICU1

**Fig. 3 Knockdown of MICU1, UCP2, and OPA1 are influencing mitochondrial cristae membrane distribution. a** TEM images of mitochondria of HeLa cells transfected with Control, MICU1, OPA1, or UCP2 siRNA. Mitochondria are highlighted in green and cristae are highlighted in magenta. **b** The amount of CM was quantified as the perimeter of cristae and normalized to the respective mitochondrial perimeter per cell. **c** Cristae density was calculated by normalizing the cristae perimeter to the mitochondrial area. **d** Schematic illustration of the measurement of spatial cristae density within mitochondria. After segmentation of mitochondria and the cristae, iterative measurements of the cristae membrane density in gradually downsized circular segments were conducted. **e** TEM images of mitochondria of HeLa cells transfected with Control siRNA (Control si) or siRNA against MICU1 (MICU1 si) or OPA1 (OPA1 si) were analyzed in regard to the spatial cristae density ($\rho_{CM}$) with the methods as displayed in **d**. On the y-axis 0 represents the most outer shell of the mitochondrion while 100 represents the mitochondrial center. Data are shown as the mean $+/-$ 95% Confidence interval. **f** For better illustration the delta cristae density ($\Delta\rho_{CM}$) to control was calculated for MICU1 and OPA1 knockdown cells. $\Delta\rho_{CM}$ to control siRNA-treated cells of MICU1 and OPA1-depleted cells were plotted in relation to the relative distance from the IBM. Data information: data in **b**, **c** are shown as dot plots representing individual cells with the mean $+/-$ SEM as middle line and whiskers, respectively ($n_{Control\ si} = 35$, $n_{MICU1\ si} = 26$, $n_{OPA1\ si} = 25$, $n_{UCP2\ si} = 30$) (**a–c**). Data in **e**, **f** are shown as mean $+/-$ SEM of individual mitochondria ($n_{Control\ si} = 3/35/82$, $n_{MICU1\ si} = 2/26/91$, $n_{OPA1\ si} = 2/26/123$, $n_{UCP2\ si} = 2/30/73$ with preparations/cells/mitochondria). For **f**, the SEM was determined by Gaussian error propagation. *$P < 0.05$ vs. respective control conditions carried out with one-way analysis of variance (ANOVA) with Bonferroni post hoc test.

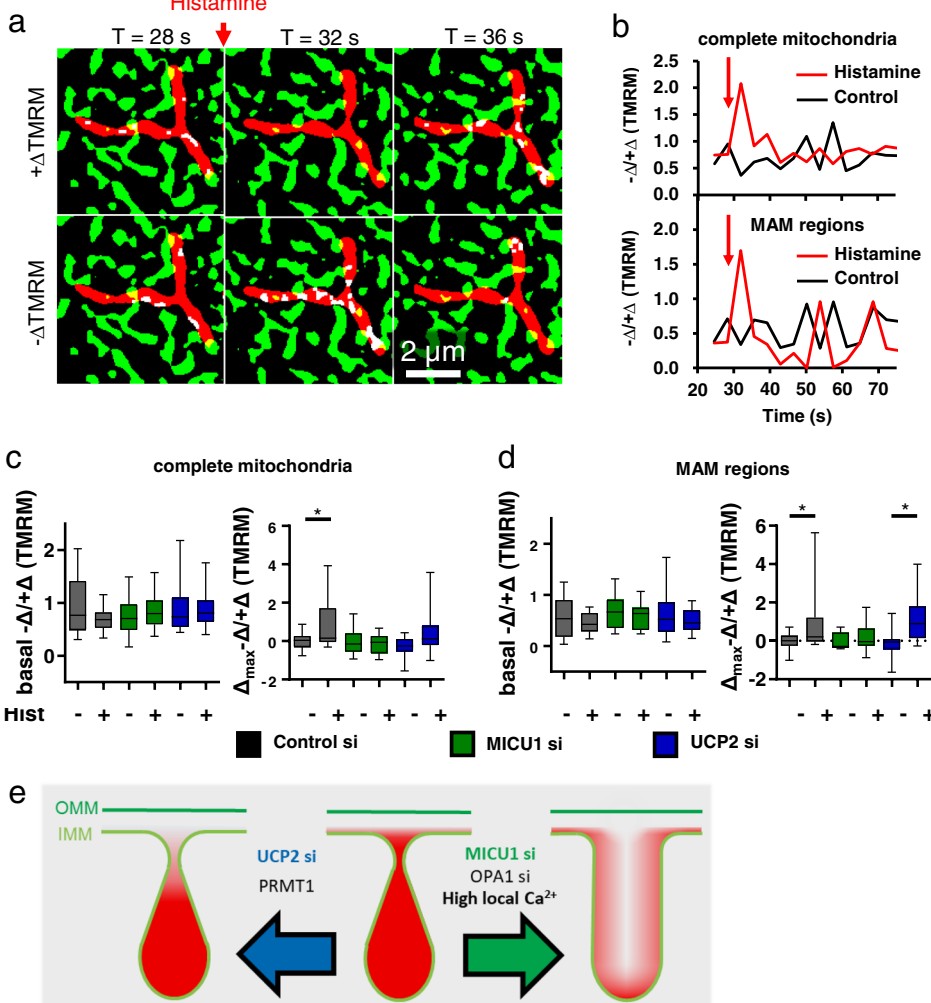

**Fig. 4 Spatial loss of $\Delta\Psi_m$ after ER Ca²⁺ release is modulated by MICU1 and UCP2. a** Binarized labeling of HeLa cells at $T = 28$ s, 32 s, and 36 s showing thresholded mitochondrial (TMRM; red), endoplasmic reticulum (ERAT 4.0 NA; green) and MAMs (yellow) structures as well as positive ($+\Delta$TMRM) or negative ($-\Delta$TMRM) local TMRM intensity changes (white). **b** Quantification of spatial restricted drops in membrane potential defined as $-\Delta/+\Delta$TMRM over time in HeLa cells challenged with or without histamine in the complete mitochondria and (upper panel) in spatial proximity to MAMs (lower panel). The red arrows point to the addition of histamine. Statistical quantification in whole mitochondria in **c** and regions with close proximity to MAMs in **d** of basal (left panel) and maximum delta (right panel) $-\Delta/+\Delta$TMRM ratios in HeLa cell transfected with siRNA against control, MICU1, or UCP2. **e** Schematic depiction of CJ opening or closing and redistribution of local membrane potential upon knockdown of UCP2, MICU1, or OPA1 as well as the influence of PRMT1 expression and high local Ca²⁺ concentrations. Data information: data are shown as the mean $+/-$ SEM ($n_{Control\ si,\ control} = 9/23$, $n_{Control\ si,\ hist} = 10/29$, $n_{MICU1\ si,\ control} = 8/24$, $n_{MICU1\ si,\ hist} = 9/26$, $n_{UCP2\ si,\ control} = 8/20$, $n_{UCP2\ si,\ hist} = 9/24$, with days/cells). *$P < 0.05$ vs. respective untreated (CaB) control conditions carried out with unpaired double-sided $T$ test.

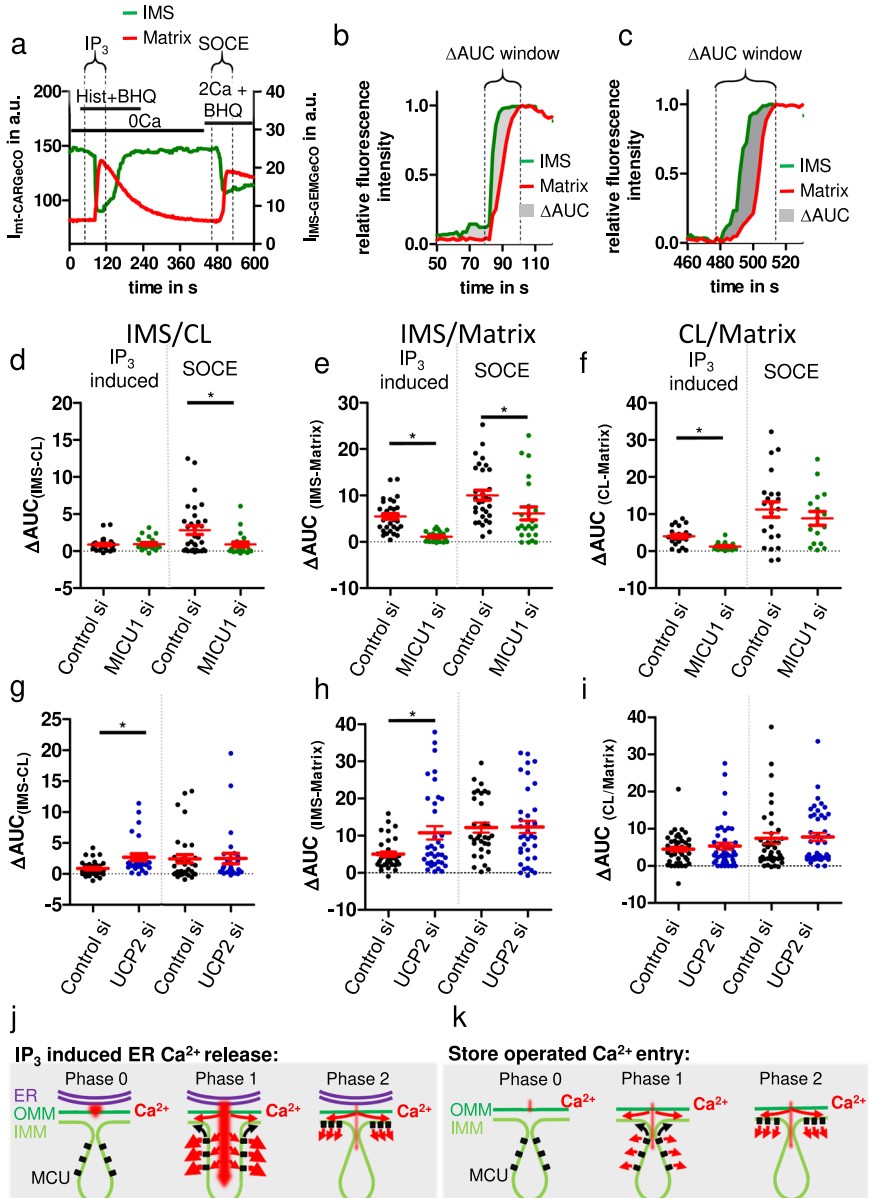

**Fig. 5 MICU1 and UCP2 modulate sub-mitochondrial $Ca^{2+}$ transition velocity. a** Single fluorescence intensities over time of IMS-GEMGeCO1 (green) and mt-CARGeCO1 (red) in HeLa cells upon stimulation with 100 μM histamine and 15 μM BHQ in a nominal $Ca^{2+}$-free buffer and subsequent readdition of 2 mM extracellular $Ca^{2+}$. **b** Normalized IMS-GEMGeCO1 (green) and mt-CARGeCO1 (red) $IP_3$ induced $Ca^{2+}$ signals with indicated ΔAUC area (gray) with upper and lower time limiting the AUC interval. **c** Normalized IMS-GEMGeCO1 (green) and mt-CARGeCO1 (red) SOCE induced $Ca^{2+}$ signals with indicated ΔAUC area (gray) with AUC interval. ΔAUC was measured in HeLa cells treated with control or MICU1 siRNA between **d** IMS-GEMGeCO1 and CL-CARGeCO1 ($n_{IP3, Control si} = 32$; $n_{IP3, MICU1 si} = 21$; $n_{SOCE, Control si} = 32$; $n_{SOCE, MICU1 si} = 21$), **e** IMS-GEMGeCO1 and mt-CARGeCO1 ($n_{IP3, Control si} = 31$; $n_{IP3, MICU1 si} = 23$; $n_{SOCE, Control si} = 31$; $n_{SOCE, MICU1 si} = 23$), and **f** CL-GEMGeCO1 and mt-CARGeCO1 ($n_{IP3, Control si} = 22$; $n_{IP3, MICU1 si} = 17$; $n_{SOCE, Control si} = 22$; $n_{SOCE, MICU1 si} = 16$) for IP3 and SOCE induced mitochondrial $Ca^{2+}$ uptake. ΔAUC was measured in HeLa cells treated with control or UCP2 siRNA between IMS-GEMGeCO1 and CL-CARGeCO1 **g** ($n_{IP3, Control si} = 34$; $n_{IP3,UCP2si} = 28$; $n_{SOCE, Control si} = 34$; $n_{SOCE, UCP2si} = 28$), IMS-GEMGeCO1 and mt-CARGeCO1 **h** ($n_{IP3, Control si} = 33$; $n_{IP3, UCP2si} = 36$; $n_{SOCE, Control si} = 33$; $n_{SOCE, UCP2si} = 36$), and CL-GEMGeCO1 and mt-CARGeCO1 **i** ($n_{IP3, Control si} = 43$; $n_{IP3, UCP2si} = 44$ $n_{SOCE, Control si} = 43$; $n_{SOCE, UCP2 si} = 44$) for IP3 and SOCE induced mitochondrial $Ca^{2+}$ uptake. **j** IP3 induced ER-$Ca^{2+}$ release leads to a biphasic $Ca^{2+}$ uptake. (1) $Ca^{2+}$ hotspot open the CJ and facilitate $Ca^{2+}$ flux into the cristae from where the $Ca^{2+}$ is taken up by constantly active MCU. (2) MCU shuttles MICU1 dependent into the IBM and enables $Ca^{2+}$ flux into the matrix circumventing the CL. **k** Store operated $Ca^{2+}$ entry is not opening the CJ due to the lack of $Ca^{2+}$ hotspots in MAMs. Accordingly, the first phase of $Ca^{2+}$ uptake is hampered resulting in a delayed $Ca^{2+}$ uptake primarily through MCU shuttling into the IBM. Data information: data are shown as scatter plots with each dot representing one cell. The mean +/− SEM are indicated as red line and whiskers. *$P < 0.05$ vs. respective control conditions carried out with unpaired double-sided $T$ test.

($EC_{50} = 4.0$ μM)[15], and *pseudo*-normalizes basal CJ $H^+$-permeability. Consistently, in the present work, depletion of UCP2 resulted in an increased CJ stability due to the low $Ca^{2+}$ binding affinity of methylated MICU1 that cannot be activated by basal

ER-originated $Ca^{2+}$ cycling within the MAMs in the absence of UCP2. In line with our findings, UCP2 knockdown is reported to increase $\Delta\Psi_m$[34], a phenomenon that, based on our present data, can be attributed to a stabilization of the CJ.

Under the condition of cell stimulation that triggers IP$_3$-induced intracellular Ca$^{2+}$ release, high Ca$^{2+}$ at the mitochondrial surface binds to MICU1 yielding its deoligomerization, subsequent opening of the CJ, and disruption of individual SMPGs. Notably, the Ca$^{2+}$ concentrations achieved within the Ca$^{2+}$ hotspots in MAMs (up to 16.4 μM) during cell stimulation[16], have the potential to disassemble methylated MICU1 (EC$_{50}$ for Ca$^{2+}$ 18.5 μM)[16] even in the absence of UCP2. Studies focusing on cristae dynamics revealed that CJ opening is predominantly restricted to MAMs after ER Ca$^{2+}$ release[30] and that CJs are closely associated with MAMs[35]. In line with this assumption, Gerencer and Vizi showed that initial mitochondrial Ca$^{2+}$ uptake after IP$_3$-induced ER Ca$^{2+}$ release originates from local foci into the whole mitochondria[36].

Recently we reported that a knockdown of MCU and EMRE leads to an enhanced accumulation of Ca$^{2+}$ inside the CL in response to IP$_3$-induced Ca$^{2+}$ release[26]. Further, the disruption of the CJ by knockdown of MICU1 or OPA1 increases basal Ca$^{2+}$ inside the mitochondrial matrix[2]. Considering both findings, one may conclude that MCU is constantly active in the CM. On the other hand, MICU1 homo and MICU1/MICU2 heterodimers were shown to increase MCU conductance[37,38] but do not occlude the MCU channel[38]. Further, three different quaternary structures including MICU1 are known : MICU1 hexamers or oligomers that disassemble Ca$^{2+}$ dependent into MICU1 dimers[14,39] and the MCU super complex involving MCU, EMRE, MICU1 and MICU2[40–42]. MICU1 was shown to potentiate MCU channel activity by direct interaction and MCU is not occluded by MICU1[38], raising the question of how MICU1 reduces mitochondrial Ca$^{2+}$ uptake at low cytosolic Ca$^{2+}$ concentrations. These findings raise the question how MICU1 is regulating both, Ca$^{2+}$ availability in the CM and amplification of the conductance of MCU? Accordingly, based on the present findings, a 2-phase model of the orchestration of MCU-mediated mitochondrial Ca$^{2+}$ uptake that meets all reported features is proposed (Fig. 5j, k):

Phase 1: *High Ca$^{2+}$ phase* with Ca$^{2+}$ hotspots instantly after IP$_3$-induced ER Ca$^{2+}$ release:

Phase 1.1. Upon IP$_3$-induced ER-Ca$^{2+}$ release and the subsequent formation of Ca$^{2+}$ hotspots within the MAMs, high Ca$^{2+}$ locally disrupts CJ by deoligomerization of MICU1.

Phase 1.2. Subsequently, the CJ opens, and Ca$^{2+}$ floats into the CL and reaches the mitochondrial matrix via constantly active MCU[38,42] in the CM.

Phase 2: *Consolidation phase* for mitochondrial Ca$^{2+}$ sequestration at the entire surface of the mitochondria, independently of the Ca$^{2+}$ source:

Phase 2.1. MCU shuttling to the IBM and assembly with MICU1-dimers or MICU1-MICU2-heterodimers[2,39,43]. Importantly, the contact with MICU1/2 increases MCU conductance and further introduces MICU1/2-mediated modulation of MCU activity that is crucial to meet the Ca$^{2+}$ micro-environment[38]. Further, the V-shaped dimeric structure of MCU, EMRE, MICU1, and MICU2 complexes is favorable for concave membranes, potentially increasing the distribution of MCU out of the convex cristae into the IBM[41,42].

Phase 2.2. Secondary to the fast transfer of ER-derived Ca$^{2+}$ to the mitochondria, this phase is important to manage the uptake of Ca$^{2+}$ entering the cells via store-operated Ca$^{2+}$ influx pathway (SOCE) that does not generate Ca$^{2+}$ hotspots in MAMs. Accordingly, SOCE-originated mitochondrial Ca$^{2+}$ uptake always occurs independently from UCP2[44].

Such sophisticated organization of the Ca$^{2+}$-triggered opening of the CJ and biphasic mitochondrial Ca$^{2+}$ uptake ensures full mitochondrial functionality even in phases of high Ca$^{2+}$ challenges or to achieve uptake of low Ca$^{2+}$. Due to local and temporal isolated CM depolarization, the mitochondria maintain general stability during

Ca$^{2+}$ uptake while increasing the metabolic output. By the local regulation of CJ opening, not only Ca$^{2+}$ uptake is regulated but also the overall ΔΨ$_{CM}$, mostly distanced from MAMs, is protected, and only affected in a minor way. This mechanism ensures mitochondrial stability under physiological Ca$^{2+}$ signaling.

Taken together, a model evolves by combining the separation of the mitochondrial membrane potential and the mitochondrial Ca$^{2+}$ uptake regulation by MICU1 at the structural level of the CJ (Figs. 4e and 5j, k). Both aspects are playing hand in hand, to fine-tune mitochondrial Ca$^{2+}$ and structural integrity to avoid matrix Ca$^{2+}$ deprivation or uncontrolled H$^+$ uncoupling through the CJ[19]. These findings have a profound influence on the understanding of mitochondrial metabolic regulation and how Ca$^{2+}$ signals are decoded into cellular responses.

## Methods

### Structured illumination microscopy (SIM)

*Single and dual camera SIM imaging.* The SIM setup used is composed of a 405, 488, 515, 532, and a 561 nm excitation laser introduced at the back focal plane inside the SIM box with a multimodal optical fiber. For super-resolution, a CFI SR Apochromat TIRF ×100-oil (NA 1.49) objective was mounted on a Nikon-Structured Illumination Microscopy (N-SIM®, Nikon, Austria) System with standard wide field and SIM filter sets and equipped with two Andor iXon3® EMCCD cameras mounted to a Two Camera Imaging Adapter (Nikon Austria, Vienna, Austria). At the bottom port, a third CCD camera (CoolSNAP HQ2, Photometrics, Tucson, USA) is mounted for wide-field imaging. For calibration and reconstruction of SIM images, the Nikon software (NIS-Elements, Nikon, Austria) was used. Reconstruction was permanently performed with the same robust setting to avoid artifact generation and ensures reproducibility with a small loss of resolution of 10% compared to the most sensitive and rigorous reconstruction settings. Microscopy setup adjustments were done as described elsewhere[2].

*Cell culture.* HeLa (ATCC-CCL-2.2TM) cells were seeded on 1.5H high precision glass cover slips (Marienfeld-Superior, Lauda-Königshofen, Germany) and cultured in DMEM (D5523, Sigma-Aldrich, Darmstadt, Germany) containing 10% FCS, penicillin (100 U/ml), streptomycin (100 μg/ml) and amphotericin B (1.25 μg/ml) (Gibco™, Thermo Fisher Scientific, Vienna, Austria) in a humidified incubator (37 °C, 5% CO$_2$/95% air). The origin of cells was confirmed via STR-profiling by the cell culture facility of the Center of Medical Research (ZMF, Graz, Austria).

*Transfection procedures.* HeLa cells were grown under standard culture conditions until 50% confluence was reached, transfected in DMEM (without FCS and antibiotics) with 1.5 μg/ml plasmids or 100 nM siRNA using 2.5 μg/ml TransFast™ transfection reagent (Promega, Madison, WI, USA). After 24 h, the medium was replaced with DMEM containing 10% FCS and 1% penicillin/streptomycin and kept for a further 24 h prior to experiments. The specific siRNAs (Microsynth, Balgach, Switzerland) used in this study are listed in Supplementary Table 1. Alternatively, plasmid transfection was done using PolyJet™ In Vitro DNA Transfection Reagent (SL100688, SignaGen® Laboratories, Frederick, MD, USA).

*Labeling with MitoTracker™ Green FM and tetramethylrhodamine methyl ester.* Cells were washed once with loading buffer containing in mM: 2 CaCl$_2$, 135 NaCl, 5 KCl, 1 MgCl$_2$, 1 HEPES, 2.6 NaHCO$_3$, 0.44 KH$_2$PO$_4$, 0.34 Na$_2$HPO$_4$, 10 D-glucose (Carl Roth, Karlsruhe, Germany), 0.1% vitamins, 0.2% essential amino acids and 1% penicillin/streptomycin at pH 7.4. Cells were incubated in loading buffer containing 1000, 500, 200, 81, 40.5, 13.5, 5.4, 2.7, or 1.35 nM TMRM (tetramethylrhodamine methyl ester, Invitrogen™) for 30 min. As TMRM might degrade over time in storage, TMRM concentrations were measured regularly. Therefore, after TMRM was dissolved in methanol the absorption at 550 nm was measured and the concentration was calculated using Eq. 1.

$$c = A/(\varepsilon \cdot l) \qquad (1)$$

$c$ is the molar concentration of TMRM, $A$ the absorption of TMRM at 550 nm, $\varepsilon$ is the specific extinction coefficient of TMRM at 550 nm in methanol, and $l$ is the pathlength the light has to path through the TMRM solution.

### Determination of TMRM quenching at various concentration

HeLa cells seeded on glass slides were labeled for 30 min with 1000, 500, 200, 81, 13.5, or 1.35 nM TMRM and transferred into a perfusion chamber. During the measurements cells were continuously perfused using a gravity-based perfusion system (NGFI, Graz, Austria) Measurements were performed on an inverted wide-field microscope (Observer.A1, Carl Zeiss GmbH, Vienna, Austria) as described previously[15]. TMRM was excited at 550 nm, and the emission was collected at 600 nm. A full disruption of the mitochondrial membrane potential was done by the application of carbonyl cyanide-p-trifluoromethoxyphenylhydrazone (FCCP) (Abcam, Cambridge, UK). In the case of the non-quenching mode of TMRM, a direct drop of

mitochondrial fluorescence followed FCCP treatment. In the case of quenching mode, a clear increase in TMRM fluorescence after FCCP addition was observed, followed by a drop in intensity. During the entire experiment the respective TMRM concentrations were present within the imaging buffers and prior recording the cells were equilibrated in the perfusion chamber for a further 5 min. Mitochondrial TMRM signals of single cells were background corrected using a background ROI.

**Determination of TMRM saturation at various concentration**. HeLa cells seeded on glass slides were labeled for 30 min with 81, 40.5, 13.5, 5.4, 2.7, or 1.35 nM TMRM and transferred into a perfusion chamber containing the same concentration of TMRM. Per coverslip, 40 cells were imaged randomly and the average cellular TMRM fluorescence was measured.

*Cristae membrane and inner boundary membrane potential separation by FWHM.* HeLa cells transfected with mt-sfGFP were stained with different concentration of TMRM ranging from 1.35 to 81 nM. After the staining procedure the cells were kept in loading buffer containing the respective concentrations of TMRM and imaged with dual-color 3D-SIM. To compensate for intensity differences for TMRM acquisition the laser power was adjusted to match image intensity histograms of different TMRM concentrations. Post imaging, the recorded data were background corrected using an ImageJ Plugin (Mosaic Suite, background subtractor, NIH) with a sliding rectangle diameter of 50 pixel. Intensity line plots of mitochondrial mt-sfGPF and TMRM fluorescence were manually measured with a width of 50 pixels (1.6 µm). The FWHM of mt-sfGFP and TMRM fluorescence distributions was measured using linear interpolation to gain subpixel information. Subtraction of $FWHM_{mt-sfGFP}$ from $FWHM_{TMRM}$ results in ΔFWHM.

*Cristae membrane and inner boundary membrane potential separation by IBM association index.* The IBM association factor of TMRM or MTG was calculated as described elsewhere[2]. In short, images were subjected to background subtraction (Mosaic Suite, background subtractor, NIH) with a sliding rectangle diameter of 50 pixels. The reference channel (sf-GFP, mtDsRed, or MTG) was Otsu[45] auto thresholded and further dilated and eroded in two independent subsets. One erosion and two dilation iterations were used. Pixel-wise subtraction of the erosion reference of the dilated reference image yields a hollow structure, used as a mask to measure the mean intensity in the mitochondrial periphery or IBM-related area in the object channel. The erosion reference served as a mask to measure the bulk or cristae mean fluorescence intensity. The ratio of IBM/CM mean intensity is a value to estimate changes in the object label distribution inside a mitochondrion, which is referred to as the IBM association index. The higher the ratio value the higher the distribution of protein labels in the IBM. For image analysis the freeware program ImageJ was used.

*Morphological analysis of mitochondria in 2D-SIM images.* 3D-SIM and time-lapsed images of EMRE-mCherry, MTR, MTG, or MICU1-YFP were used for morphological analysis. Images were background corrected with an ImageJ Plugin (Mosaic Suite, background subtractor, NIH) and the subsequent binarization was done using an Otsu[45] auto threshold. The ImageJ particle analyzer was used to extract the mitochondrial count (*c*), area (*a*), perimeter (*p*), minor (*x*), and major (*y*) axes of the mitochondria. Aspect ratio (AR) was determined as

$$AR = \frac{y}{x} \qquad (2)$$

The Form Factor (FF) was determined as followed:

$$FF = \frac{p^2}{4\pi \cdot a} \qquad (3)$$

*Electron microscopy.* Cells were incubated in a loading buffer with 81, 13.5, or 1.35 nM TMRM for 30 min, washed with PBS, fixed with 2.5% glutardialdehyde and 2% formaldehyde in a buffered solution, and postfixed in either 2% osmium tetroxide or 1% osmium tetroxide that had been reduced with 1% potassium hexacyanoferrate[46]. The cells were dehydrated in an ascending ethanol series, embedded in TAAB embedding resin, and sectioned on a Leica Ultracut 7 ultramicrotome using a Diatome diamond knife. The sections were counter-stained using platinum blue (IBIlabs, Boca Raton, FL, USA) and lead citrate (Leica Microsystems, Wetzlar, Germany) and visualized in an FEI Tecnai 20 transmission electron microscope. They were photographed at ×2000 magnification with a Gatan ultrascan 1000 camera.

*Analysis of cristae membrane and density.* In a first step, mitochondria were segmented inside the micrographs manually by free hand selection. Afterwards, the segmented mitochondria were isolated and the cristae inside each mitochondrion was segmented by free hand selection in ImageJ. The process was semi-automated using ImageJ macros. Mitochondrial and cristae area and perimeter were measured. The perimeter of mitochondria divided by the cristae perimeter gives an indication of membrane alteration of the cristae. The ratio of cristae perimeter divided by mitochondrial area gives a representation of cristae density.

*Analysis of cristae density distribution.* For analysis of the spatial cristae density distribution, segmented mitochondria and respective cristae were selected.

Segmented mitochondria were binarized and used as a mask for binarized cristae membranes to determine the percentage coverage of cristae perimeter inside the mitochondria area. Iteratively, the mitochondrial mask was eroded homogeneously in small increments of 2-pixel width (5.88 nm) and the respective coverage of cristae was measured. The result of these measurements represented a cristae density in circular segments starting from the outer mitochondrial membrane towards the mitochondrial center. As mitochondrial size and shape vary that results in different numbers of ring segments, the cristae densities were normalized by linear interpolation to 100 segments.

*Analysis of local drops in membrane potential in close proximity to MAMs.* Cells were transfected with an insensitive ER targeted ATP probe ERAT4.03 NA targeted ER (NGFI, Graz, Austria[47,48]) as a marker for the ER and stained with 13.5 nM TMRM in loading buffer for 30 min. TMRM was present during the entire experiment. Cells were afterwards transferred to the live cell chamber containing 0.8 ml loading buffer and imaged over time with a frequency of 0.25 Hz. 8 s after image sequence initialization 0.8 ml of loading buffer with or without 200 µM histamine was added to a final concentration of 100 µM histamine.

To analyze whether local drops of membrane potential appear under SOCE, cells were incubated with 2 µM Fura-2AM (MoBiTec GmbH, Göttingen, Germany) and 13.5 nM TMRM in loading buffer for 30 min. TMRM was present during the entire experiment. Cells were challenged in $Ca^{2+}$-free buffer with 100 µM histamine/10 µM CPA (Sigma-Aldrich, Vienna, Austria) to empty ER-$Ca^{2+}$ stores and imaged using the widefield setup with 380 nm illumination and widefield acquisition with the bottom port CCD camera. Afterward, the same cells were imaged over time with a frequency of 0.25 Hz using the dual-cam SIM setup. 8 s after image sequence initialization $Ca^{2+}$ was added to a final concentration of 2 mM.

A custom-made ImageJ macro was used to analyze spatial drops in mitochondrial membrane potential in dependency of spatial proximity to ER-related mitochondrial-associated membranes (MAM) and ER $Ca^{2+}$ release. The analysis was separated into several steps, (1) definition of MAMs, (2) identification of drops in the mitochondrial membrane potential, and (3) spatial assignment of the drops to MAM and non-MAM mitochondrial areas.

(1) After acquisition and reconstruction using NIS-Elements AR the mitochondrial TMRM channel (further referred to as $C_M$) and the ERAT4.03 NA channel (further referred to as $C_E$) were background subtracted (Mosaic Suite, background subtractor, NIH) with a sliding rectangle diameter of 50 pixels. Both channels ($C_M$ and $C_E$) were bleaching corrected using the histogram matching function, which is implemented in Fiji. An Otsu auto threshold was used to binarize $C_M$ and $C_E$. The overlap of both channels was eroded (iteration = 1; count = 1) to remove small structures and subsequently dilated (iteration = 6; count = 1) and used as a mask for $C_M$ to assign areas of mitochondria with close proximity to MAMs.

(2) Background subtracted (Mosaic Suite, background subtractor, 50 pixels, NIH) and bleaching corrected (histogram matching) $C_M$ was analyzed for frame-to-frame intensity changes ($\Delta C_M$). Additionally, Otsu auto thresholded $C_M$ of both images involved for the calculation of $\Delta C_M$ was used as a mask for $\Delta C_M$ to remove the mitochondrial movement from $\Delta C_M$. Negative changes from one frame to the next ($-\Delta C_M$) as well as positive ones ($+\Delta C_M$) were subjected to a threshold of 35% of the original intensity of the first frame. Thresholded $-\Delta C_M$ and $+\Delta C_M$ were eroded (iteration = 1; count = 1) and subsequently dilated (iteration = 1; count = 1) to remove remaining halos around mitochondria still originating form mitochondrial movement. Finally, the area of $-\Delta C_M$ and $+\Delta C_M$ containing segments were quantified and the ratio of $-\Delta C_M/+\Delta C_M$ was calculated. For more easy interpretation, $-\Delta C_M$ and $+\Delta C_M$ were replaced with $-\Delta TMRM$ and $+\Delta TMRM$, respectively. The area of $-\Delta C_M$ quantifies distinct local losses of TMRM signal, which can be induced by direct loss of membrane potential or by the movement of the cristae membrane itself. Shifts in the lateral or axial direction within mitochondria can lead to a loss of TMRM fluorescence. Nevertheless, cristae membrane movement should account for increases in TMRM signals in the same quantity. A ratio of $-\Delta/+\Delta TMRM = 1$ states that equal amounts of distinct changes, increase or loss of TMRM signal, are present and can be referred to a steady state in which movement of cristae membrane is the main contribution to $-\Delta TMRM$ and $+\Delta TMRM$. $-\Delta/+\Delta TMRM < 1$ shows more increases in TMRM signal indicating the generation of mitochondrial membrane potential. $-\Delta/+\Delta TMRM > 1$ on the other side implies a local high loss of membrane potential.

The whole mitochondrial area derived from auto Otsu binarized CM as well as MAMs determined in (1) were used as a mask for $-\Delta TMRM$ and $+\Delta TMRM$ areas to differentiate $-\Delta/+\Delta TMRM$-ratios in regard to the proximity to MAMs.

*Technical note.* Lateral and axial movements of the mitochondria and the CM over time may cause errors in the readings of the intensiometric TMRM signals. To counter this issue, the delta TMRM fluorescence ($\Delta C_M$) of subsequent frames was calculated and thresholded by ± 35% of the fluorescence intensity change per frame and per pixel and classified them into positive ($+\Delta TMRM$) and negative

($-\Delta$TMRM) area partitions. Accordingly, the ratio of $-\Delta/+\Delta$TMRM compensates for errors due to CM movements and quantifies local variations of membrane potential of the IMM. The ER-marker was used to identify mitochondrial structures in close proximity to the ER (MAMs) (Fig. 4a). The ratio of $-\Delta/+\Delta$TMRM was calculated over time.

**$Ca^{2+}$ imaging experiments**. $Ca^{2+}$ imaging was performed as described elsewhere[49]. In brief, $Ca^{2+}$ was measured on a digital wide-field microscope, the iMIC (Till Photonics, Gräfelfing, Germany) equipped with a 40-objective (alpha Plan Fluar 40, Zeiss, Göttingen, Germany) and an ultrafast switching monochromator, the Polychrome V (Till Photonics). Illumination of GEM-GeCO1 targeted sensors was performed at 430 nm excitation and emissions were collected with a dichrotome dual emission filter set (dichroic 535dcxr). For dual recordings, GEM- and CAR-GeCO1 targeted sensors were alternately excited for 500 ms each at 430 and 575 nm. Emissions derived from both sensors were taken in 3 s intervals. During the measurements, cells were continuously perfused using a gravity-based perfusion system (NGFI, Graz, Austria) and images were recorded with a charge-coupled device (CCD) camera (AVT Stingray F-145B, Allied Vision Technologies, Stadtroda, Germany). Data acquisition and control of the digital fluorescence microscope were performed using the live acquisition software version 2.0.0.12 (Till Photonics). For analysis, fluorescent traces of single cells were background corrected using a background ROI, and bleaching was corrected using an exponential decay function in a costume made excel macro. To compare the kinetics of two mitochondrial compartments, the fluorescent traces were normalized to the minimum and maximum of the $IP_3$ or SOCE-induced mitochondrial $Ca^{2+}$ elevations. The area under the curve was determined for both compartments using the trapezium method. The delta of both AUCs was calculated. The starting point of the $\Delta$AUC was set by the point at which the $Ca^{2+}$ signal in one of both compartments surpasses the three-fold of the standard deviation of the basal signal. The end of the $\Delta$AUC was set to the time point at which one of the signals reached the maximum and crosses with the intensity of the opposite $Ca^{2+}$ trace from the other sub-mitochondrial compartment.

**Western blot**. Western blots were performed according to standard protocols. Briefly, cell lysis was conducted with RIPA buffer (Bio-Rad formulation) supplemented with protease inhibitor cocktail (1:50; Sigma Aldrich, Vienna, Austria), followed by sonication (80% amplitude, $2 \times 15$ s). Samples were denatured in 1× Laemmli sample buffer and resolved on a 7.5 or 12.5% SDS-PAGE gel together with PageRuler™ Plus Prestained Protein Ladder (Fisher Scientific, Vienna Austria). Blots were blocked and antibodies diluted in 5% BSA (Sigma Aldrich) in TBS-T. The following antibodies were used: MICU1 (D4P8Q, 1:1000, Cell Signaling Technology, MA, USA), and Histone H3 (1B1B2, 1:1000, Cell Signaling Technology). HRP labeled anti-mouse (PI-2000, 1:1000, Vector Laboratories, Burlingame, USA) and anti-rabbit (sc-2357, 1:1000, Santa Cruz Biotechnologies) were used as secondary antibodies. For visualization, the SuperSignal West Pico PLUS kit (Fisher Scientific) was used and detection was conducted on the ChemiDoc System (Bio-Rad Laboratories, Vienna, Austria).

**Statistics and reproducibility**. Each exact $n$ value and the number of independent experiments are indicated in each figure legend. Statistical analysis was performed using the GraphPad Prism software version 5.04 (GraphPad Software, San Diego, CA, USA) or Microsoft Excel (Microsoft Office 2013). Analysis of variance (ANOVA) with Bonferroni post hoc test and $t$ test were used for evaluation of the statistical significance. All box plots show minimum to maximum values if not otherwise indicated. The central line is the median with boxes extending to 25 and 75% and the whiskers encompass all data. $P < 0.05$ was defined to be significant. At least three experiments on 3 different days were performed for each experimental set-up.

**Reporting summary**. Further information on research design is available in the Nature Research Reporting Summary linked to this article.

## Data availability
The data that support the findings of this study are available from the authors on reasonable request, see author contributions for specific data sets.

## Code availability
The code used in this study is available from the authors on reasonable request; see "Author contributions" for specific code sets.

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

## Acknowledgements

The authors wish to thank René Rost, Dr.rer.nat. and Anna Schreilechner, BSc and Luca A. Schmid for their excellent technical assistance. This work was supported by the Austrian Science Fund (FWF) (DK-MCD W1226 to W.F.G., P28529, and I3716 to R.M.), the MEFO Graz (to W.F.G.), and Nikon Austria (to W.F.G.). B.G. is supported by Nikon Austria. Z.K., F.E.O., M.H., and O.A.B. are doctoral fellows in the doctoral program Metabolic and Cardiovascular Disease (MCD) (FWF, DKplus W 1226-B18) at the Medical University of Graz. S.R. is a fellow of the doctoral program Molecular Medicine (MolMed) at the Medical University of Graz. The SIM equipment is part of the Nikon Center of Excellence, Graz that is supported by the Austrian infrastructure program 2013/ 2014, Nikon Austria Inc. and BioTechMed.

## Author contributions

B.G., Z.K., F.E.O., M.H., O.A.B., and M.W.-W. performed experimental work and FRET measurements, M.W.-W. cloned the constructs, B.G., G.L., and S.R. planned and performed the electron microscopy experiments, B.G. performed super-resolution microscopy and image data analyses and created Figs. 1b, 3d, 4e, and 5j, k. W.F.G. together with R.M. supervised the research and project planning, performed data interpretation, and prepared the manuscript. All authors discussed the results and implications and commented on the manuscript at all stages.

## Competing interests

The authors declare no competing interests.
