## [Peer Review File · Communications Biology]

REVIEWERS' COMMENTS:

Reviewer #1 (Remarks to the Author):

My concerns have been addressed in this revised manuscript.

Reviewer #2 (Remarks to the Author):

The authors reasonably addressed my original concerns.

Reviewer #3 (Remarks to the Author):

My previous concerns were fully addressed. Thank you, no further comments.

1 Response to the referees: We are grateful for the referees' valuable and insightful comments and their efforts to provide us valuable help for improving our work. Accordingly, we have addressed each point raised by the referees as follows: Reviewer # 1: This manuscript addresses the physiological roles of the MICU1 protein, which has been shown to be a regulatory subunit of the mitochondrial calcium uniporter complex. Using super-resolution microscopy and EM, it was proposed that MICU1 has a fundamental function of regulating mitochondrial inner membrane potential gradients (MPG). The finding is interesting, but I have significant concerns about assay validation and incorporation of latest results from the literature. First of all, the super-resolution imaging in Figs. 1 and 2 does not seem to really resolve the cristae structure. In Wolf et al., (EMBO J., 2019), the cristae structure is very clear and is shown to be perpendicular to the long axis of the mitochondrion as would be expected. However, the staining in Figs. 1 & 2 is quite irregular, raising concerns about whether it represents cristae. The authors should improve their imaging methods, and verify their cristae imaging at the minimum by applying their methods to cell lines known to have disrupted cristae structure. Thank you very much for your comments. We are certain that the cristea structures visible in the figures 1 and 2 illustrate cristae structures, which are not perpendicular to the focal plane. As shown in multiple 3D-EM reconstructions 1–3 . Perpendicular lamellar cristae structures are part of the ultrastructural organization of the mitochondrial cristae but not the predominate structural arrangement. According our own experiences, images shown by Wolf et al. are possible and achievable but are not representative. Further, to ensure comparable reconstruction quality in between different TMRM concentrations rather conservative reconstruction parameters were used. This was necessary because different TMRM concentrations result in different fluorescent proton yields using the same exposure and gain settings. The laser intensity was adapted to the brightness of the samples. Despite these efforts, intensity differences especially in the low TMRM range would have led to insufficient reconstruction quality if we would have used more stringent reconstruction parameters. To demonstrate that our approach is valid under our experimental conditions, we provide images of swollen mitochondria induced by Ca²⁺ overload with low or not visible internal cristae structures in the new supplementary figures 3 (Result section line 108 to line 112). Notably, the approach chosen is not dependent on visualizing and delimiting single cristae structures but takes advantage of the cross-section intensity distribution, which is much more reliable to measure with the resolution provided by structured illumination microscopy. Second, the MPG assay was not validated. In Wolf et al., the dependence of TMRE patterns on $\Delta\Psi$ was rigorously verified by flickering. In addition to flickering, I would also recommend adding FCCP to test if this eliminates the MPG. The measurement of MPG is rather new, and a standard approach has not been established in the field. Without careful validation of the cristae imaging and the voltage dependence of TMRM stain, I am not convinced that the authors' interpretation of the results is meaningful. 2 Thank you very much for this important advice. Like TMRE, TMRM is a validated membrane potential dependent dye accumulating in the inner mitochondrial membrane^{4,5}. TMRM is used in a wide field of publications to measure the mitochondrial membrane potential ⁶ . We added the respective citations in line 88. We frequently use FCCP to normalize the readouts of the membrane dyes, but this intervention entirely removes the mitochondrial specific staining (see supplementary figures 2). Thus $\Delta FWHM$ and AIBM cannot be calculated. However, the sensitivity and strict dependency of TMRM distribution/localization is demonstrated in OPA1 knockdown cells. In control cells, TMRM accumulates preferentially in the cristae. In contrast, if cristae junction is opened by OPA1 knockdown, TMRM, even at low concentrations, is homogeneously distributed over the entire IMM. These data demonstrate that TMRM is suitable to monitor membrane differences between the IBM and cristae. Similar experiments with

OPA1 silencing were done by Wolf et al. to verify the analysis of different membrane potentials between CM and IBM7. Finally, the MICU1 hexamer structure was only observed in the first MICU1 structure paper by Wang et al. Later structures, including the 3 full MCU complex structures published in the last year, demonstrate that MICU1 forms a dimer. There does not seem to be a Ca²⁺-dependent reorganization of MICU1 oligomers as claimed here. Such discrepancy should be addressed. Thank you for this important point. Both, MICU1 hexamer and MICU1-MCU complexes were published and are both in agreement as well as needed for the mechanism postulated in this paper. In the paper of Wang et al. exclusively MICU1 and its derivatives were used for structural analysis⁸, while other recent papers used a mixture of MCU, EMRE, MICU1, and MICU2 to analyze the MCU supercomplex^{9,10}. Both approaches have their justification and both, MICU1 hexamers as well as MCU supercomplexes, are included in the mechanism proposed in this manuscript. The hexameric MICU1 structure disassembles under high Ca²⁺ and forms the MCU complex together with MCU and EMRE. Hexameric MICU1 and MCU supercomplex structures might differ from the real structure, as membrane interactions and the high IMM curvature, especially in the cristae junction, might affect the quaternary structure of these multi-protein complexes. Further, we have recently determined the influence of Ca²⁺ on MICU1 deoligomerization^{11–13} using a FRET approach in intact cells. These results show clearly the dependency of MICU1 oligomerization on IMS Ca²⁺ levels in intact cells. To point to the aspects raised by the referee, we added a respective part in the discussion section to implement the different molecular structures of MICU1-hexamer and MCU complex into our model (line 302 to line 307 and line 321 to line 323).

3 Reviewer #2: The manuscript by Gottschalk et al investigates local differences in membrane potential in cristae vs IBM and its dependence on local Ca²⁺ dynamics. Overall, the paper is potentially interesting, but additional work is needed to solve some issues, at both methodological and conceptual levels. Methods: • Here, the authors claim to have established “a reliable and robust method microscopy to detect spatial membrane potential gradients using super-resolution microscopy”. However, no real validation for this method is shown. For instance, “we assumed that at a given high concentration TMRM gets saturated in the CM while the TMRM accumulation maintains in a linear correlation in the IBM with its lower $\Delta\Psi$ ”. I think there is no need to “assume” this point, but rather this can be experimentally verified. Please also note that TMRM signal is not “saturated” as reported, but rather its fluorescence undergoes to self-quenching. Accordingly, ref 21 has little to do with quenching/dequenching mode of TMRM. We thank the reviewer for that interesting and very important point. We used concentrations that are far from usual quenching mode measurements using 400 – 1000 nM TMRM^{14–16}. We also measured TMRM fluorescence in our cell model and could verify that within concentration of 1.35 to 81 nM no TMRM self-quenching can be observed. 200 nM TMRM showed a tendency but no significant results for TMRM self-quenching. We added these results in Supplementary Figure 2. Accordingly, we don’t think that quenching mode of TMRM is applicable in our study. To further address the point raised by the referee, we tested if TMRM accumulation in HeLa cells follows a non-linear correlation with the used TMRM concentrations. We found that TMRM is not following a concentration dependent linear accumulation after 30 min of incubation, but rather a saturation kinetic with higher concentration. We added these results as new Supplementary Figure 1. • Please consider that other super resolution techniques (e.g. PMID: 31609012) showed heterogeneity among different cristae within the same mitochondrion. Is this approach able to detect the same features? Is it possible to distinguish membrane potential of individual cristae along with differences in CM vs IBM potentials? Thank you very much for this valuable comment. However, this approach was designed to detect variations of the mitochondrial membrane potential along the cross section of mitochondria, but not to

differentiate between single cristae invaginations. The optical resolution for this approach would have to match or be smaller than inter cristae distances of less than 60 nm to differentiate between single cristae convincingly. Accordingly, we measured mitochondrial cross sections over a larger mitochondrial area including several cristae to acquire an average of the membrane potential distribution. 4 • Quantification of spatial distribution of CM (Fig 3d-f) is also less than ideal. To my understanding, the authors analyzed “consecutive circular segments of mitochondria”, but the large majority of mitochondria are not circular in this samples, based on both representative EM images (Fig. 3a) and quantitative analysis (according to fig S3, aspect ratio is approximately 3, thus far from being “circular”). In addition, please consider that silencing of MICU1 and OPA1 causes fragmentation, while UCP2 knockdown causes elongation. Most likely, this will increase/decrease the circularity of mitochondrial, potentially causing artifacts not related to CM distribution. We thank the reviewer for this important point. We defined the OMM of the mitochondria as a mask. That mask was consecutively eroded homogeneously by a defined amount (See Methods section line 437 - 445). Accordingly, the mitochondrial shape does not influence the measurement, as the cristae area or membrane is measured at the smallest distance to the OMM independent of the mitochondrial shape. As we think the illustration of figure 3d was misleading for the reviewer not showing the exact methodology, we replaced the scheme with a more accurate new one in which the homogeneous erosion of the mitochondrial mask is shown more clearly. To prevent misunderstandings, we include in the old and new scheme here. new old For illustration, we added two mitochondria with different shape and the incremental measurement borders of the masks created. Shown are the incrementally shrunk masks of Control (right) and UCP2si (left) of figure 3a. Please, be aware that the shown increments are only a subset of the actually measured increments (see line 441 and 444). 5 Results: • It is not clear what is the contribution of changes in global organelle morphology vs specific changes in cristae structure. Indeed, Fig S3 clearly shows that the various genetic interventions significantly affect global mitochondrial morphology. In particular, silencing of either OPA1 or MICU1, causes mitochondrial fragmentation (lower form factor and aspect ratio, shorter major axis). Conversely, silencing of UCP2 triggers mitochondrial elongation (higher form factor and AR, longer major axis) and increased CJ density (Fig S4). However, EM images should in principle have enough resolution to quantify CJ, thus opening the possibility to correlate data on membrane potential (global vs local potential) with specific structural parameters. Indeed we have exactly performed such a correlation in our recent paper¹³. We found that the CJ is widened through knockdown of MICU1 or OPA1, correlating with reduced overall membrane potential and increased basal mitochondrial Ca²⁺ levels. We assume that the IMM constriction and mitochondrial Ca²⁺ increase which precedes mitochondrial fission in ER close regions¹⁷, supports a model in which Ca²⁺ hotspots¹⁸ in MAMs lead to MICU1 deoligomerization, the opening of the CJ (Figure 4a-e), constriction of the IMM, and finally to mitochondrial fission. Thus, a reduction of CJ stabilizing proteins like MICU1 or OPA1 leads to a higher rate of mitochondrial fission and altered mitochondrial morphology. We added a respective paragraph in the discussion We added a paragraph regarding that issue in the discussion section (Line 265 – 270) • It would be nice to support data on the contribution of MICU1 using mutants in the EF domains. Indeed, silencing of MICU1 has been shown to increase resting matrix [Ca²⁺], maybe with an impact on local extramitochondrial [Ca²⁺] As pointed out by the referee, MICU1 silencing indeed increases resting matrix Ca²⁺ concentration as also we have frequently reported. However, MICU1-EF-hand mutants do not undergo structural rearrangement during ER Ca²⁺ release¹¹, because the multimerization site is around the EF hands. Further, we could not see any changes in mitochondrial morphology in cells expressing MICU1-EF-hand mutants¹³. Wang

et al. showed that EF-mutants do not rescue reduced mitochondrial Ca²⁺ uptake caused by silencing of endogenous MICU1. These findings and our own results suggest that EF-mutants do not undergo Ca²⁺-induced rearrangement and block the CJ for Ca²⁺ transit. Indeed, silencing of MICU1 might reduce Ca²⁺ concentrations in "Ca²⁺ hotspots" between the ER and mitochondria because of an easier Ca²⁺ transit into the organelle below the K_d of MICU1. • Please provide also representative traces of Ca²⁺ transients in different sub-compartments. We added representative traces of all mitochondrial sub-compartments as new supplementary figure 12.

Reviewer #3: Gottschalk et al used SIM, TEM, mitochondria membrane potential and potential and mitochondria targeted Ca sensors to understand the role of MICU1 in cristae junction and its effects on mitochondria Ca uptake. The authors conclude that Ca released from the ER binds to MICU1 disrupting CJ barrier that allows Ca to enter cristae and thereby to be rapidly absorbed into the matrix by constitutively active MCU. Establishment of "CJ barrier" function of MICU1 using nontrivial elegant approaches is new. However, some points require clarification to adequately support the conclusions. Major: 1) Timing of the processes: The authors demonstrate that manipulation with MICU1, OPA1 or UCP2 expression affect mitochondria cristae diameter and membrane potential in this compartment. Further, the authors suggest that ER-Ca release affect MICU1-dependent cristae junction opening. However, there are no dynamic measurements directly visualizing cristae opening upon ER Ca release (and closing when release ceases). One could suggest alternative explanation for the results obtained: i.e.: MICU1 directly controls MCU activity. MICU1 KD increases matrix Ca uptake which results in dissipation of mitochondria membrane potential etc. UCP2 KD on the other hand may reduce uptake (PMID: 20403634), thereby no change in membrane potential upon MICU1 KD... It is plausible that the loss of membrane potential gradient and CJ barrier is secondary to disturbed mitochondria Ca handling. Can you completely block mitochondria Ca uptake to test whether it plays (or not playing) a role in cristae opening and membrane potential depolarization? Thank you very much for this interesting point. Direct visualization of CJ opening after ER Ca²⁺ release is a highly challenging task. It would require a dynamic system combined with very high resolution equally powerful like electron microscopy. Here we can only provide indirect measurements and basics, published previously by us and others. MCU independent CJ opening in MICU1 silenced cells (CJ width measured with EM) 13, MICU1 structural rearrangement measured using a FRET approach 11, the localization of MICU1 to the IBM 13, its interaction with the MICOS-complex 19, altered Ca²⁺ uptake kinetics (this paper), homogeneous membrane potential distribution (this paper), MICU1 dependent local membrane potential fluctuations in ER-close regions (this paper), and the involvement of MICU1 in the kinetics of cristae, particularly in ER-close regions 20, all point to a Ca²⁺ dependent role of MICU1 in the stability of the CJ. A recent publication by Garg et al. showed that MCU channel activity is not occluded by MICU1. Instead, MICU1/2 potentiates the activity of MCU as extramitochondrial Ca²⁺ is elevated 21, leading to the conclusion that inhibition of MCU by MICU1 cannot originate from a direct interaction of MCU and MICU1 in the reported MCU/EMRE/MICU1/MICU2 complex.. 2). The authors show changes in mitochondria structure in cells transfected with siRNAs. It implies that mitochondria function is also affected, which may disrupt ER Ca release. One may suspect that changes in mitochondria Ca kinetics upon ER Ca release activation may be caused largely by differences in ER Ca release and not by switching mitochondria Ca uptake modes. It would be beneficial to perform more quantitative measurements of Ca release amplitudes and mitochondria [Ca] using F_{min}/F_{max} normalization procedure. Literature shows that knockdown of MICU1 22, OPA1 23 or UCP2 24,25 does not change cytosolic Ca²⁺ transients. Intensiometric quantitative analysis of IP3 and SOCE induced 7 mitochondrial Ca²⁺ signals are shown in the supplementary figure 11 and originate from the same data

as our Δ AUC measurements. 3) Is it possible to visualize MCU translocation from cristae? We recently published a paper focusing on the redistribution of MCU from the cristae into the IBM dependent on IMS Ca^{2+} and MICU1 13. Minor: Could you please present western blot data showing siRNAs efficiency? Thank you for mentioning that important point. We did verify the knockdown efficiency of the siRNAs used against UCP2 (PCR) 12 (Western blot) 25, OPA1 (PCR)20 (Western blot) 20, PRMT1 (PCR)12 (Western blot) 12 and MICU1 (PCR)26 elsewhere. We added the respective citations (Lines 124, 137 and 144) into the manuscript and added Western blots showing the knockdown efficiency of siRNA against MICU1 to the manuscript in supplementary figure 4. References 1. Noh, Y. H. et al. Inhibition of oxidative stress by coenzyme Q10 increases mitochondrial mass and improves bioenergetic function in optic nerve head astrocytes. *Cell death & disease* 4, e820; 10.1038/cddis.2013.341 (2013). 2. Stephan, T. et al. MICOS assembly controls mitochondrial inner membrane remodeling and crista junction redistribution to mediate cristae formation. *The EMBO journal* 39, e104105; 10.15252/embj.2019104105 (2020). 3. Strubbe-Rivera, J. O. et al. Modeling the Effects of Calcium Overload on Mitochondrial Ultrastructural Remodeling. *Applied sciences (Basel, Switzerland)* 11; 10.3390/app11052071 (2021). 4. Ishigaki, M. et al. STED super-resolution imaging of mitochondria labeled with TMRM in living cells. *Mitochondrion* 28, 79–87; 10.1016/j.mito.2016.03.009 (2016). 5. Kondadi, A. K. et al. Cristae undergo continuous cycles of membrane remodelling in a MICOS-dependent manner. *EMBO reports* 21, e49776; 10.15252/embr.201949776 (2020). 6. Perry, S. W., Norman, J. P., Barbieri, J., Brown, E. B. & Gelbard, H. A. Mitochondrial membrane potential probes and the proton gradient: a practical usage guide. *BioTechniques* 50, 98–115; 10.2144/000113610 (2011). 7. Wolf, D. M. et al. Individual cristae within the same mitochondrion display different membrane potentials and are functionally independent. *The EMBO journal* 38, e101056; 10.15252/embj.2018101056 (2019). 8. Wang, L. et al. Structural and mechanistic insights into MICU1 regulation of mitochondrial calcium uptake. *The EMBO journal* 33, 594–604; 10.1002/embj.201386523 (2014). 9. Zhuo, W. et al. Structure of intact human MCU supercomplex with the auxiliary MICU subunits. *Protein & cell* 12, 220–229; 10.1007/s13238-020-00776-w (2021). 10. Fan, M. et al. Structure and mechanism of the mitochondrial Ca^{2+} uniporter holocomplex. *Nature* 582, 129–133; 10.1038/s41586-020-2309-6 (2020). 11. Waldeck-Weiermair, M. et al. Rearrangement of MICU1 multimers for activation of MCU is solely controlled by cytosolic Ca^{2+} . *Scientific reports* 5, 15602; 10.1038/srep15602 (2015). 12. Madreiter-Sokolowski, C. T. et al. PRMT1-mediated methylation of MICU1 determines the UCP2/3 dependency of mitochondrial Ca^{2+} uptake in immortalized cells. *Nature communications* 7, 12897; 10.1038/ncomms12897 (2016). 13. Gottschalk, B. et al. MICU1 controls cristae junction and spatially anchors mitochondrial Ca^{2+} uniporter complex. *Nature communications* 10, 3732; 10.1038/s41467-019-11692-x (2019). 14. *Mitochondria*, 2nd Edition (Elsevier, 2007). 15. Gerencser, A. A. et al. Quantitative measurement of mitochondrial membrane potential in cultured cells: calcium-induced de- and hyperpolarization of neuronal mitochondria. *The Journal of physiology* 590, 2845–2871; 10.1113/jphysiol.2012.228387 (2012). 16. Lemasters, J. J. & Ramshesh, V. K. Imaging of Mitochondrial Polarization and Depolarization with Cationic Fluorophores. In *Mitochondria*, 2nd Edition (Elsevier 2007), Vol. 80, pp. 283–295. 17. Cho, B. et al. Constriction of the mitochondrial inner compartment is a priming event for mitochondrial division. *Nature communications* 8, 15754; 10.1038/ncomms15754 (2017). 18. Giacomello, M. et al. Ca^{2+} hot spots on the mitochondrial surface are generated by Ca^{2+} mobilization from stores, but not by activation of store-operated Ca^{2+} channels. *Molecular cell* 38, 280–290; 10.1016/j.molcel.2010.04.003 (2010). 19. Tomar, D. et al. MICU1 regulates mitochondrial cristae structure and function independent of the mitochondrial calcium uniporter channel (2019). 20. Gottschalk, B., Klec, C., Waldeck-Weiermair, M., Malli, R. & Graier, W. F.

Intracellular Ca²⁺ release decelerates mitochondrial cristae dynamics within the junctions to the endoplasmic reticulum. *Pflügers Archiv : European journal of physiology* 470, 1193–1203; 10.1007/s00424-018-2133-0 (2018). 21. Garg, V. et al. The mechanism of MICU-dependent gating of the mitochondrial Ca²⁺ uniporter. *eLife* 10; 10.7554/eLife.69312 (2021). 22. Mallilankaraman, K. et al. MICU1 is an essential gatekeeper for MCU-mediated mitochondrial Ca²⁺ uptake that regulates cell survival. *Cell* 151, 630–644; 10.1016/j.cell.2012.10.011 (2012). 23. Fülöp, L., Szanda, G., Enyedi, B., Várnai, P. & Spät, A. The effect of OPA1 on mitochondrial Ca²⁺ signaling. *PloS one* 6, e25199; 10.1371/journal.pone.0025199 (2011). 24. Graier, W. F., Trenker, M. & Malli, R. Mitochondrial Ca²⁺, the secret behind the function of uncoupling proteins 2 and 3? *Cell calcium* 44, 36–50; 10.1016/j.ceca.2008.01.001 (2008). 25. Trenker, M., Malli, R., Fertschai, I., Levak-Frank, S. & Graier, W. F. Uncoupling proteins 2 and 3 are fundamental for mitochondrial Ca²⁺ uniport. *Nature cell biology* 9, 445–452; 10.1038/ncb1556 (2007). 26. Waldeck-Weiermair, M. et al. Leucine zipper EF hand-containing transmembrane protein 1 (Letm1) and uncoupling proteins 2 and 3 (UCP2/3) contribute to two distinct mitochondrial Ca²⁺ uptake pathways. *The Journal of biological chemistry* 286, 28444–28455; 10.1074/jbc.M111.244517 (2011).

Reviewers' comments:

Reviewer #1 (Remarks to the Author):

This manuscript addresses the physiological roles of the MICU1 protein, which has been shown to be a regulatory subunit of the mitochondrial calcium uniporter complex. Using super-resolution microscopy and EM, it was proposed that MICU1 has a fundamental function of regulating mitochondrial inner membrane potential gradients (MPG). The finding is interesting, but I have significant concerns about assay validation and incorporation of latest results from the literature.

First of all, the super-resolution imaging in Figs. 1 and 2 does not seem to really resolve the cristae structure. In Wolf et al., (EMBO J., 2019), the cristae structure is very clear and is shown to be perpendicular to the long axis of the mitochondrion as would be expected. However, the staining in Figs. 1 & 2 is quite irregular, raising concerns about whether it represents cristae. The authors should improve their imaging methods, and verify their cristae imaging at the minimum by applying their methods to cell lines known to have disrupted cristae structure.

Second, the MPG assay was not validated. In Wolf et al., the dependence of TMRE patterns on $\Delta\Psi$ was rigorously verified by flickering. In addition to flickering, I would also recommend adding FCCP to test if this eliminates the MPG. The measurement of MPG is rather new, and a standard approach has not been established in the field. Without careful validation of the cristae imaging and the voltage dependence of TMRM stain, I am not convinced that the authors' interpretation of the results is meaningful.

Finally, the MICU1 hexamer structure was only observed in the first MICU1 structure paper by Wang et al. Later structures, including the 3 full MCU complex structures published in the last year, demonstrate that MICU1 forms a dimer. There does not seem to be a Ca^{2+} -dependent reorganization of MICU1 oligomers as claimed here. Such discrepancy should be addressed.

Reviewer #2 (Remarks to the Author):

The manuscript by Gottschalk et al investigates local differences in membrane potential in cristae vs IBM and its dependence on local Ca^{2+} dynamics. Overall, the paper is potentially interesting, but additional work is needed to solve some issues, at both methodological and conceptual levels.

Methods:

- Here, the authors claim to have established "a reliable and robust method microscopy to detect spatial membrane potential gradients using super-resolution microscopy". However, no real validation for this method is shown. For instance, "we assumed that at a given high concentration TMRM gets saturated in the CM while the TMRM accumulation maintains in a linear correlation in the IBM with its lower $\Delta\Psi$ ". I think there is no need to "assume" this point, but rather this can be experimentally verified. Please also note that TMRM signal is not "saturated" as reported, but rather its fluorescence undergoes to self-quenching. Accordingly, ref 21 has little to do with quenching/dequenching mode of TMRM.
- Please consider that other super resolution techniques (e.g. PMID: 31609012) showed heterogeneity among different cristae within the same mitochondrion. Is this approach able to detect the same features? Is it possible to distinguish membrane potential of individual cristae along with differences in CM vs IBM potentials?
- Quantification of spatial distribution of CM (Fig 3d-f) is also less than ideal. To my understanding, the authors analyzed "consecutive circular segments of mitochondria", but the large majority of mitochondria are not circular in this samples, based on both representative EM images (Fig. 3a) and quantitative analysis (according to fig S3, aspect ratio is approximately 3, thus far from being "circular"). In addition, please consider that silencing of MICU1 and OPA1 causes fragmentation, while

UCP2 knockdown causes elongation. Most likely, this will increase/decrease the circularity of mitochondrial, potentially causing artifacts not related to CM distribution.

Results:

- It is not clear what is the contribution of changes in global organelle morphology vs specific changes in cristae structure. Indeed, Fig S3 clearly shows that the various genetic interventions significantly affect global mitochondrial morphology. In particular, silencing of either OPA1 or MICU1, causes mitochondrial fragmentation (lower form factor and aspect ratio, shorter major axis). Conversely, silencing of UCP2 triggers mitochondrial elongation (higher form factor and AR, longer major axis) and increased CJ density (Fig S4). However, EM images should in principle have enough resolution to quantify CJ, thus opening the possibility to correlate data on membrane potential (global vs local potential) with specific structural parameters.
- It would be nice to support data on the contribution of MICU1 using mutants in the EF domains. Indeed, silencing of MICU1 has been shown to increase resting matrix $[Ca^{2+}]$, maybe with an impact on local extramitochondrial $[Ca^{2+}]$
- Please provide also representative traces of Ca^{2+} transients in different sub compartments.